# Coastal lake sediments from Arctic Svalbard suggest colder summers are stormier

Zofia Stachowska [1], Willem G. M. van der Bilt [2] ✉ & Mateusz C. Strzelecki [3]

The Arctic is rapidly losing its sea ice cover while the region warms faster than anywhere else on Earth. As larger areas become ice-free for longer, winds strengthen and interact more with open waters. Ensuing higher waves also increase coastal erosion and flooding, threatening communities and releasing permafrost carbon. However, the future trajectory of these changes remains poorly understood as instrumental observations and geological archives remain rare and short. Here, we address this critical knowledge gap by presenting a continuous Holocene-length reconstruction of Arctic eolian activity using coastal lake sediments from Svalbard. Exposed to both polar Easterlies and Westerly storm tracks, sheltered by a bedrock barrier, and subjected to little post-glacial uplift, our study site provides a stable baseline to assess Holocene changes in the dominant wind systems of the Barents Sea region. To do so with high precision, we rely on multiple independent lines of proxy evidence for wind-blown sediment input. Our reconstructions reveal quasi-cyclic summer wind maxima during regional cold periods, and challenge the view that a warmer and less icy future Arctic will be stormier.

The Arctic responds faster to on-going climate change than any other region on Earth[1–3]. Over the past forty years, regional warming has progressed nearly four times faster than the global average[2]. This rapid transformation is most visibly manifested by a rapid decline of the Arctic's sea ice cover[4,5]. As a result, larger areas remain ice-free longer, and the fetch—the distance over which wind can interact and transfer energy to surface waters—expands[6–9]. Associated increases in wave height and frequency are further exacerbated by the vulnerability of thinning remnant ice to wind-driven fracturing[10]. While observations of Arctic wave climate are rare and mostly local, these changes have increased wave height by up to 30 cm per decade in some areas[11]. The impact of wind-driven wave energy, amplified by permafrost degradation and sea-level rise, is the main driver of coastal erosion along vast tracts of Arctic shoreline[12–18], which threatens coastal communities[16], while releasing more carbon than all of the region's rivers combined[19].

Despite the afore-mentioned environmental and socio-economic impacts, the magnitude of future changes in Arctic storminess under warmer and less icy conditions remains poorly constrained. This is well-illustrated by the divergence between climate models: while some suggest a weakening or southward shift of the mid-latitude Westerlies, and the North Atlantic storm tracks in particular[20–22], others present evidence for their consistent poleward shift[23,24]. As climate models are calibrated using observational data, their projections generally become less reliable when variability exceeds the range of instrumental records[25]. Paleoenvironmental data from geological archives are well-suited to fill this critical knowledge gap, by providing us with longer-term baseline data on the links between changes in climate and storminess under different climate conditions. Arctic coastal deposits are often well-preserved as isostatic uplift rates have typically outpaced global sea-level rise since deglaciation[26,27], potentially preserving coastal sediment sequences that cover most of the Holocene. By effectively trapping eolian particles and sea-spray aerosols[28,29], sediments from coastal lakes are prime archives to record past changes in wind strength and wave height—henceforth refered to as storminess. Critically, high-fidelity sediment core scanning techniques and

[1]Institute of Marine and Environmental Sciences, Doctoral School, University of Szczecin, Szczecin, Poland. [2]Department of Earth Science and Bjerknes Centre for Climate Research, University of Bergen, Bergen, Norway. [3]Alfred Jahn Cold Regions Research Centre, Institute of Geography and Regional Development, University of Wrocław, Wrocław, Poland. ✉e-mail: willemvanderbilt@uib.no

**Table 1 | Proxy correlations**

|  | CT | DBD | inc./coh. | LOI | log (Zr/K) | MGS | EM 1 | EM 3 |
|---|---|---|---|---|---|---|---|---|
| DBD | 0.61 |  |  |  |  |  |  |  |
| inc./coh. | −0.65 | −0.80 |  |  |  |  |  |  |
| LOI | −0.56 | −0.93 | 0.71 |  |  |  |  |  |
| log (Zr/K) | −0.49 | −0.76 | 0.71 | 0.69 |  |  |  |  |
| MGS | *-0.15* | *-0.09* | 0.28 | *-0.04* | 0.40 |  |  |  |
| EM 1 | 0.25 | *0.17* | −0.32 | *-0.09* | −0.43 | −0.89 |  |  |
| EM 3 | *-0.04* | *-0.03* | *0.21* | *-0.14* | 0.26 | 0.86 | −0.66 |  |
| Ti (TSN) | 0.67 | 0.81 | −0.93 | −0.71 | −0.87 | −0.34 | 0.37 | −0.26 |

Values reflect Spearman's correlation coefficients ($\rho$), and those spelled with italics reflect results with $p \geq 0.05$.

geochronological advances allow us to reconstruct past changes on human-relevant (decades to centuries) timescales[29]. However, while several lake sediment-based North Atlantic wind reconstructions have been published in recent years[30–32], the potential of coastal Arctic lakes to record changes in paleostorminess remains under-utilized[33].

Here, we present a continuous Holocene-length lake sediment-based paleostorminess reconstruction from Svalbard—an Arctic climate change hotspot[34]. This Archipelago is uniquely sensitive to changes in key drivers of storminess as both warming and sea ice melt rates exceed the regional average[35,36]. We analyze a ~9700 year long sediment sequence from the southern tip of Svalbard—Sørkappøya island. Protected by a bedrock barrier and exposed to little post-glacial emergence[37], our study site—coastal Lake Steinbruvatnet—provides a stable baseline to assess Holocene changes. To rigorously reconstruct storminess on multicentennial to millennial timescales, we employ a multi-proxy approach that combines independent geochemical (X-Ray Fluorescence; XRF), visual (Computed Tomography; CT), and granulometric (End-Member Modeling Analysis; EMMA) lines of evidence for wind-transported particles in a statistical (Principal Component Analysis; PCA) framework (Table 1). Our findings suggest that Holocene summertime wind maxima occurred during cold periods, and challenge the emerging notion that a warmer and less icy future Arctic will be stormier. However, more regional records are needed to confirm the representativeness of our findings.

## Results and Discussion
### Setting
Sørkappøya is a 7 km-long island off the southern tip of Spitsbergen—the biggest island of the Svalbard Archipelago (Fig. 1a, b). In contrast with other parts of the island group[37], adjacent Sørkapp Land has experienced only modest sea-level changes after deglaciation ~11,000–9000 cal. yrs B.P. as shoreline uplift has not exceeded 10 m over the last 6500 years[38,39], enhancing the preservation-potential of Holocene-length archives of coastal change (Fig. 1b, d)[37]. The bedrock is composed of Palaeozoic and Mesozoic sedimentary and low-grade metamorphic rocks[40,41]. In addition, large areas are covered by unconsolidated Quaternary deposits[40]. The coastal geomorphology of the island is dominated by three major components: (I) rocky ridges and spurs in the West, that serve as wave breakers, and often constitute the structural anchor for (II) uplifted beach ridges to the East, and (III) numerous coastal lakes separated from the sea by hooked spits and barriers (Fig. 1d and S1, and Supplementary Note 1).

Coastal lakes effectively capture the products of wave- and wind-transported input like sea-spray aerosols and minerogenic grains[28,29,31]. To harness this potential, we targeted Lake Steinbruvatnet for this study (Fig. 1c) due to its unique setting. Notably, situated close to the polar front (Fig. 1a)[42], the 2.5 m deep basin is impacted by both polar Easterlies and Westerly storm tracks (Fig. 1a, b), so that wind-blown input might derive from both systems. Indeed, the presence of sandy shadow dunes to the West of Steinbruvatnet and silt sheets to the East of the lake indicate efficient inland eolian transport of sediment

(Fig. S1b, c). The observed East-West grain size difference can be traced back to the source of mobilized material: the West coast is characterized by a high (2–4 m) gravel-dominated storm ridge perched on a rocky shore platform (Fig. 1d; Fig. S1b), while the East coast is characterized by a flatter ~50 m wide beach where many silty and sandy deposits can be found (Fig. 1d and S1c). Moreover, the lake is protected from erosion and disturbance by storm surges as it is situated 5 m above sea-level (a.s.l.) and sheltered by an 8 m a.s.l. rocky ridge to the West, as well as a 1 km wide beach ridge plain to the East (Fig. 1d). Lake Steinbruvatnet also lacks in- or outlets, and bears no evidence of lake level fluctuations, limiting the potential for non-eolian catchment-derived minerogenic input. Finally, the preservation of periglacial forms (i.e., sorted circles, ice wedge polygons) and beach ridges with gentle eolian features (i.e., dunes) suggest that the island has not been affected by storm overwash events (also see Fig. 1d). In conclusion, the catchment geology and geomorphology suggest a comparatively sheltered setting, despite its exposure to wind.

Climatologically, available wind observations measured from 2013 onwards on Sørkappøya reveal that the Easterlies dominate during wintertime (DJF), while wind directions are evenly distributed in summer (JJA)[43]. The Westerlies are, however, generally weaker as wind speeds rarely (0.5% of the time) reach gale force, whereas the Easterlies do so during on 10% of winter days[43]. Also, timeseries analysis of Sentinel-2 satellite imagery reveals that the lake is ice-covered for ~9 months per year[44]. At present, our study area on Svalbard is situated close to the rapidly retreating seasonal sea ice maximum (Fig. 1b)[45], while biomarker (IP$_{25}$) evidence suggests that seasonal sea ice only became widespread during the last millennium of the Holocene[46]. Ongoing changes are closely linked to the 1 °C per decade warming trend observed in the region[35]. Today, local mean air temperatures remain below zero at -3.7 °C, and the annual amount of precipitation averages 478 mm per year at nearby Hornsund station[47].

### Core chronology
As also shown in Table 2, 4 of the 13 radiocarbon dates taken from the core 601-21-6 GC were excluded from our core chronology as their size fell short of laboratory size requirements for precise dating. The other 9 radiocarbon dates were incorporated in an age model created with the help of version 3.2.0 of the Bacon R package[48]. Ages were calibrated with IntCal20 curve and reported with a 2 sigma (2$\sigma$) uncertainty range (cal. yrs B.P.; see Fig. 2 and Table 2)[49]. Stratigraphically inverted old ages were identified as outliers: we note that the age of all of these cluster ~10,500 cal. yrs B.P. As only terrestrial plant macro-fossils were used for this model (see the chronology paragraph in our methods section), we argue that these anomalous ages derive from reworked land deposits. In support of this evidence[39], suggest that local sea-level was up to 5 meters lower than today around this time. This is based on ~10,000 cal. yrs B.P old wave-scoured peat remains from the now-submerged 3-8 m deep embayment that separates Sørkappøya island from adjacent Sørkapp Land on mainland Spitsbergen. The same authors show that a transgression culminated in our

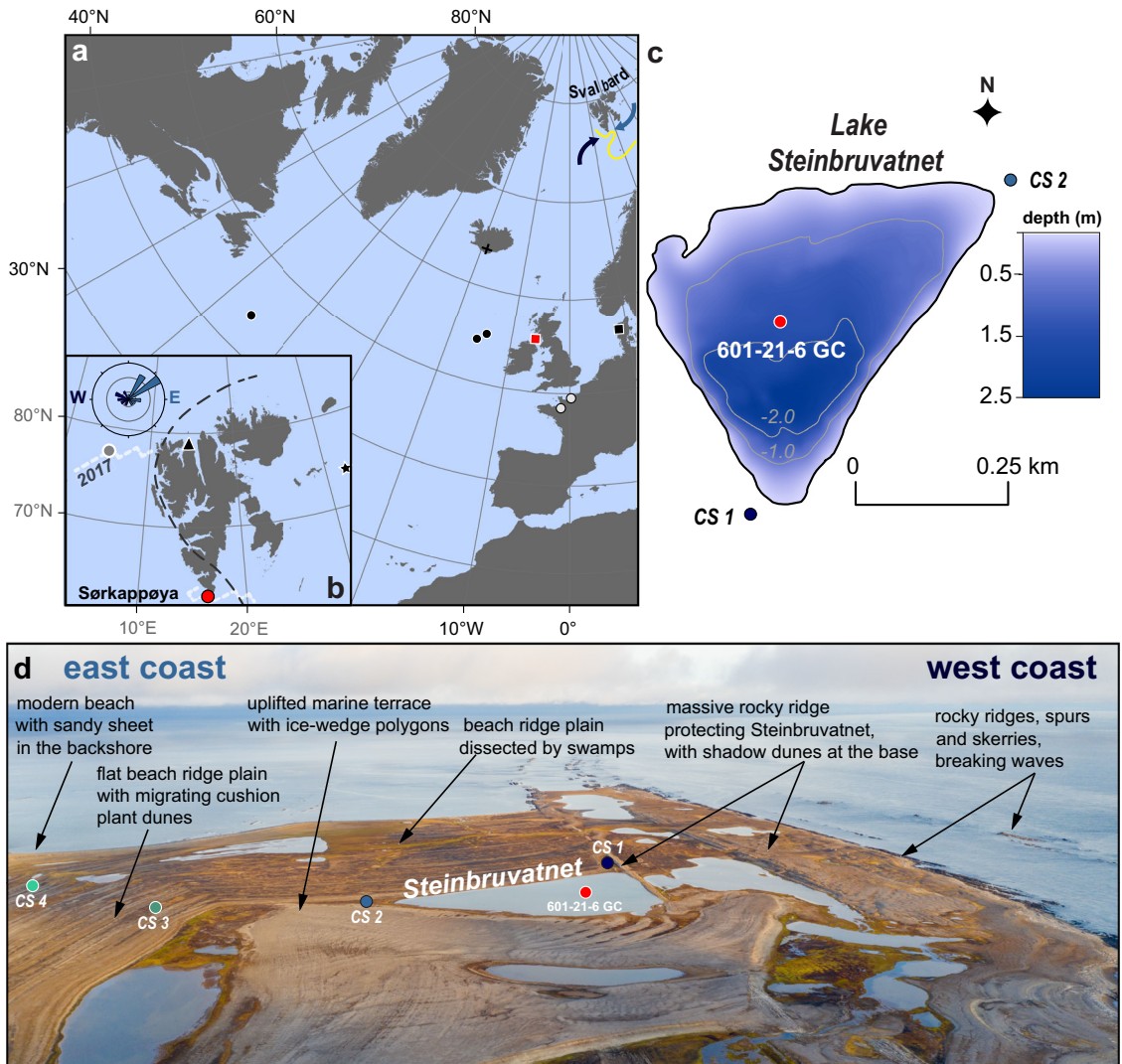

**Fig. 1 | Overview maps of the North Atlantic, Svalbard, and Sørkappøya.**
**a** Localities of key regional Holocene climate records used to contextualize our data: sand-sized soil grains deposition by polar Easterly storm events in Iceland (black plus)[65], a stacked record of Holocene Storm Periods (HSPs) from the North Atlantic (white dots)[66], Aeolian Sand Influx (ASI) to Lake Filsø in western Denmark (black square)[30], grain-size variability from the Laphroaig coastal peat bog in southwestern Scotland (red square)[58], and the Ice Rafted Debris (IRD) stack from the North Atlantic (black dots)[91,92]. The yellow line marks the modern position of the polar front after[42], where the Easterlies and Westerlies (indicated by arrows in colors that match their representation in other figures) meet. **b** Svalbard, with the location of our study area on Sørkappøya (red dot), the frequency of eolian sand-sized particles from Lake Vårfluesjøen in northern Spitsbergen (black triangle)[33], $P_BIP_{25}$-

derived sea ice coverage in the Fram Strait (gray dot)[73,93], $P_BIP_{25}$-derived ice coverage in the northern Barents Sea (black star)[70], and $U^K_{37}$-based sea surface temperature (SST) from the Barents Sea margin[71,94]. The white stippled lines indicate minimal (CE 2017) sea ice limits (March)[95], while the black dashed line shows the 0 m postglacial emergence isobase for Svalbard[37]. The wind rose shows the wind direction distribution on western Svalbard between CE 1947-2018[43]. **c** Bathymetry of Lake Steinbruvatnet with 1 m contour isobaths in gray. The location and name of the analyzed sediment core are highlighted in red. CS 1 and CS 2 indicate the source of catchment samples to the West and East of the lake, respectively. **d** Aerial imagery of the Steinbruvatnet catchment (photo by R. Stange), with the location of key geomorphological features affecting eolian transport around Lake Steinbruvatnet, and the location of catchment samples CS 1-4.

study area close to the elevation of Lake Steinbruvatnet ca. 8000 cal. yrs B.P. Based on these findings, we argue that the distinct decline in sediment accumulation rates (SARs) seen ~8000 cal. yrs B.P. (Fig. 2d) may be linked to a progressive increase in distance from the sea: as the marine processes that often supply sediments in coastal lakes like Steinbruvatnet waned[30], accumulation slowed. The rudimentary sea-level curve compiled for Sørkapp Land by Ref. 37 supports this notion. In this context, we would like to stress that neither catchment geomorphology nor lake sedimentology bear evidence of a direct marine influence on the lake, for example via storm surge events (see our setting paragraph). Regardless, both the afore-mentioned outliers as well as our terrestrial ca. 9700 cal. yrs B.P. basal age (Table 2) suggest that the Steinbruvatnet catchment was isolated from the ocean multiple millennia earlier than previously reported by Ref. 37. Finally, we

note that sedimentation rates between our uppermost radiocarbon ages are indistinguishable from the values inferred for the core top (Fig. 2d), which is constrained by the year of sediment collection – 2021 C.E. This strengthens our confidence in the presented model, despite its significant uncertainties, and furthermore suggests that sediments from the top 7 cm of our core are reworked but complete.

## The Holocene evolution of Steinbruvatnet
As outlined the previous section, visual assessment reveals that sediment from the uppermost 7 cm of the investigated Steinbruvatnet record (core 601-21-6 GC: see methods) has been homogenized (Suppl. Fig. S2). We argue that this lack of structure stems from reworking, likely due to post-coring disturbance (mixing the sediment-water interface). We therefore exclude the uppermost 7 cm from further

## Table 2 | Radiocarbon sample overview

| Lab code | Depth (cm) | Material | Dry weight (mg) | 14C age (yrs B.P.) | Error (yrs) | Cal. yrs B.P. |
|---|---|---|---|---|---|---|
| Ua-76380 | 7.5 | Terrestrial plants | 2.7 | 9312 | 45 | 10,601–10,369 |
| Poz-177580* | 9.5 | Terrestrial plants | 1 | 9580* | 60 | 11,160–10,730* |
| Poz-150330* | 11.75 | Terrestrial plants | 0.9 | 6290* | 35 | 7289–7158 |
| Ua-76381 | 15.5 | Terrestrial plants | 7.1 | 9461 | 42 | 10,790–10,574 |
| Poz-177581* | 18.5 | Terrestrial plants | 3.7 | 9780* | 60 | 11,315–11,080* |
| Poz-177582* | 18.5 | Aquatic moss | 1.9 | 6070* | 50 | 7025–6790* |
| Poz-149887 | 22.25 | Terrestrial plants | 11.8 | 3475 | 35 | 3839–3682 |
| Poz-177257 | 30.5 | Terrestrial plants | 1 | 9630 | 60 | 10,975–10,770 |
| Poz-150331 | 46.25 | Terrestrial plants | 9.7 | 6180 | 40 | 7166–6953 |
| Poz-149888 | 58.75 | Terrestrial plants | 13.7 | 7130 | 50 | 8024–7915 |
| Ua-76382 | 79.25 | Terrestrial plants | 6.0 | 7714 | 38 | 8552–8415 |
| Ua-76383 | 85.25 | Terrestrial plants | 4.3 | 9376 | 46 | 10,718–10,495 |
| Poz-145555 | 105.75 | Terrestrial plants | 36.2 | 8650 | 50 | 9742–9532 |

All ages were extracted from analyzed core 601-21-6-GC. Calibrated ages, errors, and ranges (2σ) are based on the Intcal20 curve[49]. * mark Poz samples with C masses below lab size recommendations, while gray numbers mark outliers.

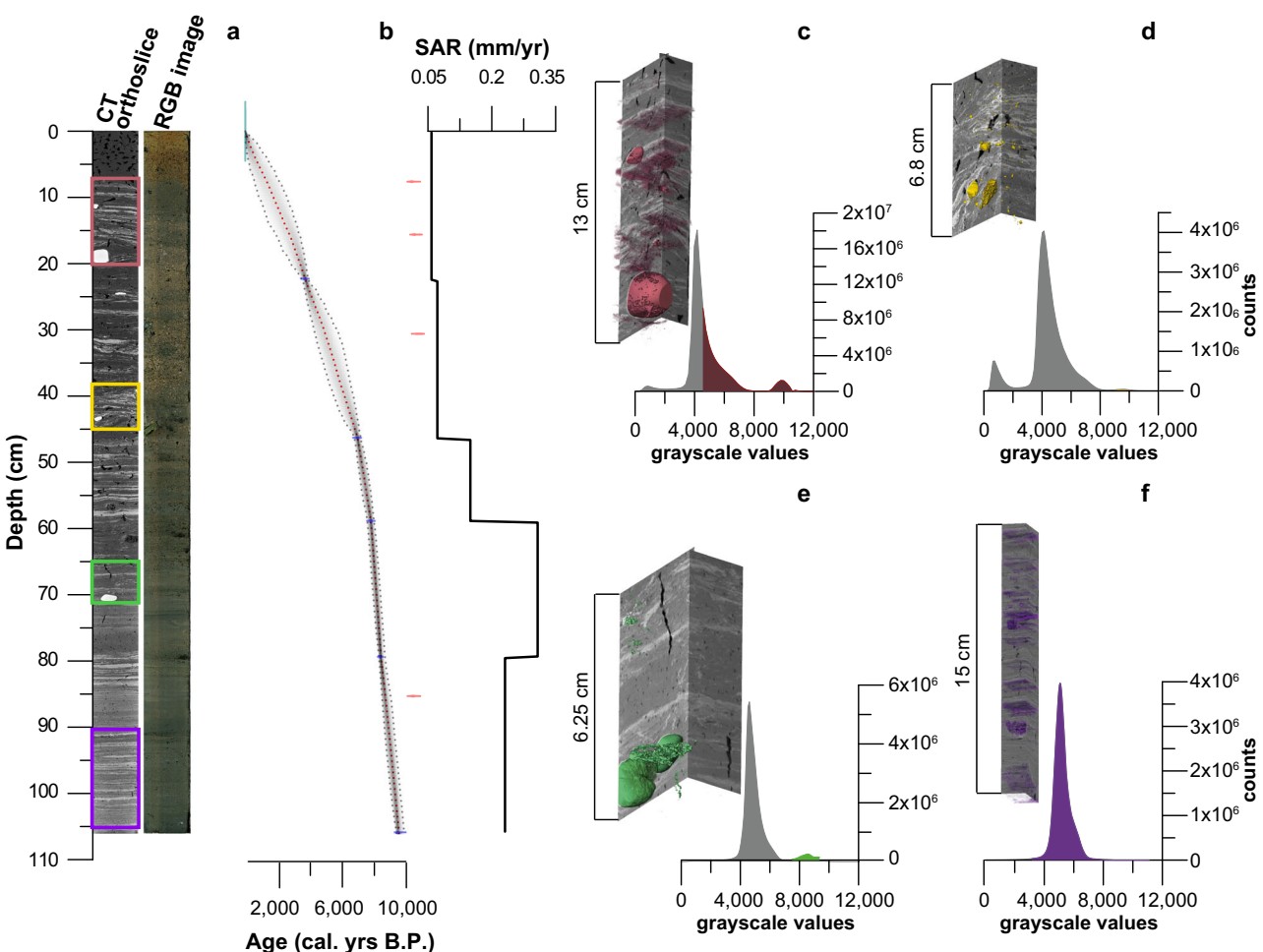

**Fig. 2 | Chronology, sedimentation rates and core visualization.** The left panel: Computed Tomography (CT) and line camera imagery of core 601-21-6 GC. **a** Our age-depth model—the red line indicates the weighted mean best fit of our model, while the gray outlines highlight its 95% confidence range. Calibrated ¹⁴C age distributions are in shown in blue (included) and red (outliers). **b** Sediment Accumulation Rates (SAR: mm/yr). **c**–**f** Characteristic dropstones and facies 2 (storm) layers in selected CT scan intervals (rectangles in matching colors). Source data for the data shown in this figure are provided as a Source Data file tab.

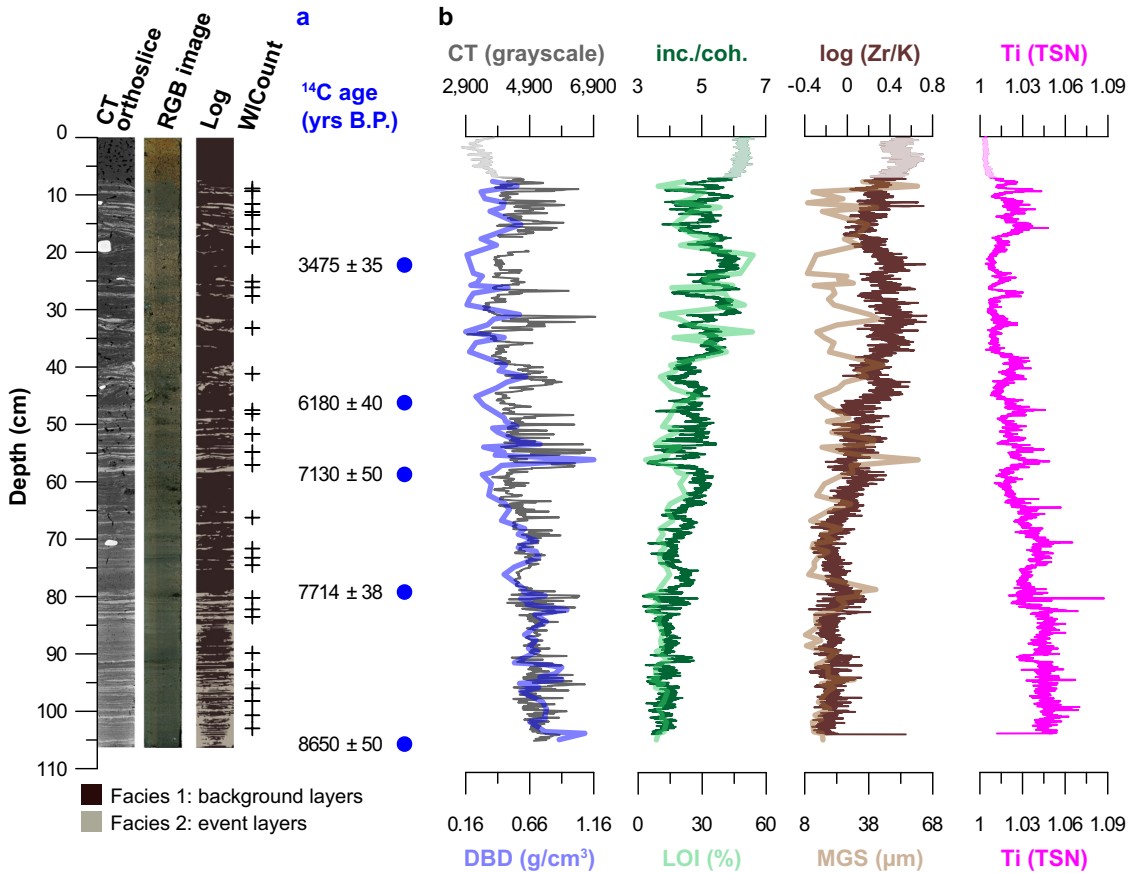

**Fig. 3 | Key proxies measured on Steinbruvatnet sediments.** The left panel: Computed Tomography (CT) and X-Ray Fluorescence (XRF) imagery of core 601-21-6 GC, a log with facies 1 and 2 highlighted, the thickest minerogenic horizons (≥1 cm thick) identified by WlCount (see Statistics paragraph in methods)[50]. **a** [14]C ages ± errors (yrs) from sampled intervals. **b** density – captured by CT grayscale values[53,54], and Dry Bulk Density (DBD), organic content – reflected by XRF incoherent/coherent (inc./coh.) scattering (top)[51,52], and Loss on Ignition (LOI) percentages, grain size variability reflected by mean grain size (MGS) values (μm) and the log ratio of Zirconium/Potassium (log (Zr/K))[29,51,62,80], and bulk minerogenic input reflected by Total Scatter Normalized (TSN) Titanium (Ti) values[29,54]. Source data for the data shown in this figure are provided as a Source Data file tab.

analysis. Following visual assessment of the record, we identify two main facies (Fig. 3a): dark brown background sediments (facies 1), and lighter-colored clastic horizons that vary in thickness from ~2 mm to ~2 cm (facies 2). The latter layers are also automatically captured by WlCount (see Statistics paragraph in our methods)[50]. Our multi-proxy analysis furthermore reveals that facies 1 is organic, as reflected by higher Loss on Ignition (LOI) and XRF incoherent/coherent (inc./coh.) ratios (Fig. 3a, c), an often-used productivity proxy[51,52]. In contrast, facies 2 layers are dense—as reflected by elevated Dry Bulk Density (DBD) and CT grayscale values[53], and minerogenic—as reflected by higher Total Scatter Normalized (TSN; see Stratigraphy paragraph in our methods section) XRF Titanium (Ti) values[29,54], a conservative element that is broadly applied as an indicator of clastic terrigenous input[51]. Ti is positively correlated with DBD and CT grayscale values ($\rho = 0.61$, $n = 96$ and $\rho = 0.67$, $n = 4729$, respectively, $p = 0.000$; Table 1). Based on the characteristics of both facies and the distinct difference between them (Figs. 2 and 3), we argue that facies 2 layers were deposited in a higher energy environment. Considering the coastal setting of investigated Lake Steinbruvatnet and the absence of in- and outlets to mobilize catchment material (see our setting paragraph), we favor an eolian origin. Moreover, although Steinbruvatnet sits just 5 m above modern sea-level, we argue that the lake has not been directly impacted by storm surges since the onset of lacustrine sedimentation ~9700 cal. yrs B.P. (see our setting paragraph), due to the absence of diagnostic features like erosive contacts between both facies, marine fossils, rip-up clasts, or event deposits in the catchment

area[55,56]. As also mentioned in our setting paragraph, this is supported by geomorphological field evidence (see Fig. 1d): the presence of well-preserved periglacial features and beach ridges preclude overwash. Finally, our record is devoid of the gravels that dominate the western beach (see our setting section, and Fig. S1b)−the likeliest source of storm surges due to its proximity to Lake Steinbruvatnet and exposure to the North Atlantic swell (Fig. 1d).

Complementing the above evidence that identifies variations in eolian input (facies 2), we explore the use of particle size distributions as a wind strength indicators after[57]. Compared to other beach-proximal storm-influenced settings[58], the silt-dominated mean grain size (MGS) distribution in Steinbruvatnet lake is comparatively fine and also stable (Fig. 3 and S3). Our End-Member Modeling Analysis (EMMA; see Stratigraphy paragraph in our methods section and Fig. S4) output provides a possible explanation[59], by demonstrating that particle size distributions are diluted with silt (End-Member 1: EM 1), which often dominate reworked glacigenic soils found in unvegetated ice-proximal polar settings like our Svalbard study area[60]. Indeed, the granulometry of a catchment sample taken from the eolian silt sheets that are found along a transect towards the East coast (CS 2-4; see Fig. 1c-d) confirm that particles transported by the polar Easterlies are dominated by this size fraction (Fig. S4).

In contrast, sand-dominated End Member (EM) 3, which also exerts an influence on mean grain size (see Table 1), has a mean much larger than anything found towards the East of Steinbruvatnet, and more similar to catchment sample CS 1−taken from an active dune to

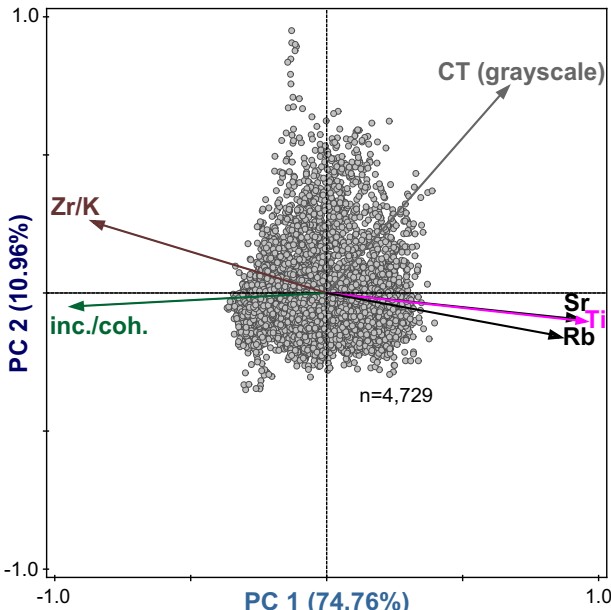

**Fig. 4 | Ordination diagram.** Sample (gray dots) and variable (arrows) scores for the two principal components (PCs) with the greatest explanatory power (labeled on axes). Source data for the data shown in this figure are provided as a Source Data file tab.

the West of the lake (see Fig. 1 and Fig. S4). We therefore argue that coarser EM 3 input was transported by the Westerly storm tracks, while attributing the grain size difference with CS 1 to winnowing with distance. Following from the above grain size evidence, we argue that the fine silt-dominated input of EM 1 is mobilized by the polar Easterlies, while coarser sand-dominated EM 3 is transported to Lake Steinbruvatnet by the Westerly storm tracks. Finally, the presented CT imagery of Figs. 2 and 3 reveal the sporadic presence of rounded pebbles and cobbles that were lake ice-rafted from the rocky western lake shore (see Fig. 1d)—which lacks finer silt and sand fractions (presumably due to wave action)—and deposited as drop stones. Between 10.4 and 12.8 cm these did not allow us to acquire CT grayscale values (Fig. 3).

To distil information from multiple of the afore-mentioned proxies that capture changes in wind regime in investigated Lake Steinbruvatnet, we relied on Principal Component Analysis (PCA; see Statistics paragraph in methods). To do so on human-relevant (decades to centuries) timescales, we decided to only include μm-scale resolution scanning data as the 0.3 cm sampling diameter of measured physical parameters exceeds the width of most facies 2 layers (see Stratigraphy paragraph in methods). We argue that this can be legitimized by the moderate to strong correlation between physical and scanning measures of organic content (LOI vs. inc./coh.), and density (DBD vs. CT), and grain size (MGS vs. log (Zr/K)) – see Table 1. As outlined in our Stratigraphy paragraph in the methods section, we excluded XRF elements with a Signal-to-Noise ratio lower than 2[54,61]. Based on the observed co-variance of our first principal component (PC 1) with minerogenic indicators Rubidium (Rb), Strontium (Sr) as well as Ti (Fig. 4), and the association of the latter element with fine-grained (EM 1-dominated) input (see Fig. 3 and Table 1), which is found along the eastern shores of Sørkappøya (see setting and Fig. S4), we associate PC 1 with eolian input from the polar Easterlies. In contrast, both particle size indicator log (Zr/K)[51,62] as well as CT grayscale values[51,62], which are also often impacted by changes in granulometry[63], have stronger PC 2 loadings (see Fig. 4). As mentioned, and shown (see Fig. S4), minerogenic input of this size fraction (EM 3-dominated) is only available on the more proximal West coast (see setting and Fig.

1d). Therefore, we argue that PC 2 tracks changes in the Westerly storm tracks. As cryogenic factors like frozen ground, ice foot, lake ice coverage, and snow cover restrict eolian transport and deposition in Arctic study areas like ours, we argue that both PC 1 and 2 capture summer-dominated wind signals. Finally, we note that the previously mentioned 10.4-12.8 cm clast-related CT data gap is also reflected in our PCA data.

### Holocene changes in Easterly and Westerly wind strength

Following from the above, we argue that the presented multi-proxy evidence from Lake Steinbruvatnet captures changes in summertime Easterly (PC 1) and Westerly (PC 2) wind strength between ca. 9700 and 1700 cal. yrs B.P. Due to the multi-centennial-scale uncertainties of our age model and a lack of undisturbed surface sediments to build modern analogs using historical storms (Fig. 2 and core chronology)[28], it is not possible to ascertain whether our PC maxima reflect windy events or phases of stronger winds. However, we favor the latter scenario after[29], as our proxies do not behave in a binary fashion and inferred eolian input dominates sedimentation throughout the record (Fig. 3). Also, although our PCs are associated with each of the wind systems that impact our study area today[43], we cannot assess absolute changes in their respective strength, due the afore-mentioned lack of observation-based validation, as well as differences in eolian particle size and transport distance (Fig. 1, S1 and S4). In addition, while the inferred changes can result from either shifts and/or strengthening of storm tracks, we cannot disentangle these components. Therefore, we will discuss our reconstructions as relative variations in Easterly and Westerly wind strength through time. To further validate our interpretations and assess their broader representativeness, we compare our records to other wind reconstructions. Such efforts are, however, often hampered by 1) data scarcity—as Holocene-length extra-tropical storm reconstructions remain scarce, 2) baseline shifts—fluxes of shore-derived eolian input are affected by sea-level changes, and 3) age uncertainty—windy phases may mis-align between sites because of chronological errors[28,64]. All these factors affect the Arctic region disproportionally due to its remoteness, complex isostatic uplift history, and a general scarcity of radiocarbon (14C) dateable material. To help overcome these challenges, we primarily focus our comparison on two regionally relevant records that resolve change on multi-centennial timescales like our PC-based wind indicators (Fig. 5), and cover most of the Holocene like our data—published reconstructions rarely extend beyond the Mid-Holocene[64]. Firstly, the only existing continuous Holocene reconstructions of the Easterly winds in the Arctic North Atlantic: locally by Ref. 33, and in Iceland by Ref. 65 (see Fig. 5). As both records are based on eolian input from local soils, they remain largely unaffected by post-glacial uplift. And secondly, the stacked chronology of past Westerly wind activity in coastal northwest Europe by Ref. 66, which identifies Holocene Storm Periods (HSPs) in nine coastal records collected from the microtidal Seine Estuary and Mont-Saint-Michel Bay in northwestern France since 6500 cal. yrs B.P., when regional sea-level change stabilized following melt of the Laurentide Ice Sheet[67]. In addition, we compare our PC data to other regionally relevant storminess records that do not meet all of the above criteria (see Fig. S5).

As outlined in the introduction, this study fundamentally seeks to deepen our understanding of the links between Arctic climate and storminess. To further highlight wind strength maxima, we compiled a so-called Storm Magnitude Index (SMI) by calculating the area under our PC curves after[54]. For this purpose, we 1) detrended PC values to negate the impact of sea-level change on coastal distance and therefore eolian material fluxes to warrant assessment against a stable baseline (see Statistics paragraph in methods) after[28], before 2) identifying extremes as peaks that exceed the mean ($\mu$) + one standard deviation ($\sigma$) bound of our PC values, and finally 3) calculating the definite integral for each of these stormy intervals using the trapezoidal rule.

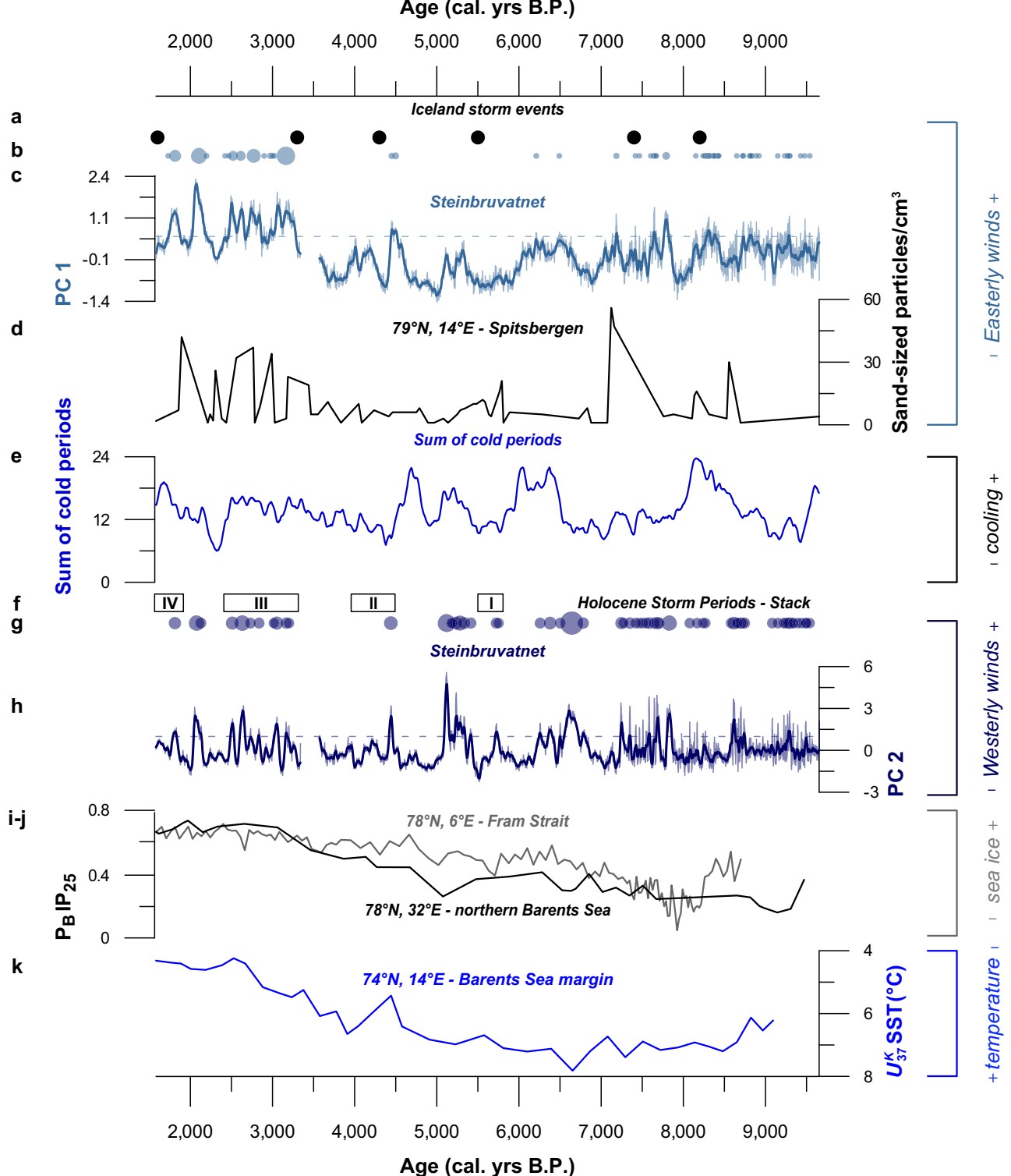

**Fig. 5 | Holocene changes in wind regime contextualized.** A comparison between our Principal Component (PC)-based Easterly and Westerly wind reconstructions and relevant reconstructions of storminess and paleoclimate data located in Fig. 1a, b using matching colors. **a** Iceland storm events by Ref. [65]. **b** PC-1 derived Storm Magnitude Index (SMI) values for polar Easterly winds, shown as scaled circles, Lake Steinbruvatnet. **c** Principal Component 1 (PC 1)-derived polar Easterlies reconstruction from Lake Steinbruvatnet, highlighting 30-year averages in bold, using a stippled line to mark the $(\mu + 1\sigma)$ cut-off for extremes. **d** The number of sand-sized particles (larger than 255 µm) from Lake Vårfluesjøen in Northern Spitsbergen[33]. **e** Sum of cold periods, based on a global set of mostly summer-biased Holocene

temperature timeseries[69]. **f** the stacked chronology of Holocene Storm Periods (HSPs) from the North Atlantic, built on core data from the Seine Estuary and Mont-Saint-Michel Bay in northwestern France[66]. **g** PC-2 derived SMI values for Westerly winds shown as scaled circles, Lake Steinbruvatnet. **h** Principal Component 2 (PC 2)-derived Westerly storm track reconstruction from Lake Steinbruvatnet, highlighting 30-year averages in bold, using a stippled line to mark the $(\mu + 1\sigma)$ cut-off for extremes. **i** $P_B IP_{25}$-derived sea ice cover data from the Fram Strait (gray), and **j** from the northern Barents Sea (black)[70,73,93]. **k** A $U^K_{37}$-based sea surface temperature (SST) reconstruction from the Barents Sea margin[71,94]. The PCA and SMI source data for the data shown in this figure are provided as a Source Data file tab.

## Regionally coherent signals of multi-centennial scale wind variability

Considering the afore-mentioned challenges that complicate comparison between sites, our PC-derived wind reconstructions from Lake Steinbruvatnet bear some resemblance to the selected regionally relevant reconstructions. As can be seen in Fig. 5 a-c, PC 1 reproduces the majority of Easterly wind events captured by Ref. 65, despite differences in signal amplitude that we tentatively attribute to **1**) location —both sites sit ~2000 km apart and therefore differently with regard to the average storm track position (Fig. 1a), **2**) proxy—the silt-sized particles associated with the Easterly winds in Steinbruvatnet lake require less wind energy for transport than the sand-sized soil grains reported by Ref. 65, and **3**) seasonality—as snow, frost and ice cover limit Arctic sediment transport and deposition, we argue that our record primarily captures summer season variability. Possibly due to similar climatic conditions and its proximity, the correspondence with the local reconstruction by Ref. 33 is greater (Fig. 5d). Similarly, as seen in Fig. 5f, g, our PC 2 SMI maxima broadly coincide with the Westerly wind HSPs reported by Ref. 66, although overlap with HSP II around 4500–4000 cal. yrs B.P. is marginal. However, when looking at our detrended PC 2 scores, we see Westerlies increase throughout this period. This difference in signal amplitude might be explained by the sensitivity of the investigated systems—the foreshore archives analyzed by Ref. 66 are more exposed to wind impacts than sheltered backshore sites like Lake Steinbruvatnet[28]. Broad correspondence and amplitude differences are also observed when comparing our data to the grain size-based Westerlies reconstructions from western Denmark[30], and the Hebrides (see Fig. S4)[58]—which is particularly similar throughout the past ~3500 yrs, when North Atlantic conditions were stormier as also noted by Ref. 68. In summary, despite site-specific differences, our Easterly and Westerly wind maxima are regionally consistent, which strengthens our confidence in their interpretation. Also, as pointed out by Ref. 66, Holocene peaks of both Easterly and Westerly wind systems coincide (Fig. 5).

## Stronger summertime winds during North Atlantic cold periods

Our PC-based reconstructions reveal regionally consistent multi-centennial Holocene maxima in between 1500-3000 cal. yrs B.P., as well as around 4500, 6500, and between 7000–9500 cal. yrs B.P (Fig. 5). As also contested by Refs. 65 and 66, stormy phases coincide with North Atlantic (summer-biased) cooling periods identified by Ref. 69 (see Fig. 5e). By extending this association between cold and windy conditions into a study area that is seasonally sea ice-covered, our findings challenge the emerging view that a warmer and less icy Arctic will become stormier—the premise of our study (see introduction). This notion is supported by local evidence, which reveals that **1**) Easterly winds were most intense during the Late Holocene when sea surface temperatures were relatively low and severe sea ice conditions persisted in the up-wind Barents Sea[70–72], while **2**) Westerly wind strength does not exhibit a clear relation with either temperature or sea ice conditions as SMI maxima occur throughout the Holocene (Figs. 1 and 5)[73]. By showing that wind input does not increase with temperature, our findings also allay concerns that a shortening of lake ice and snow cover during warmer periods significantly impacts sediment mobilization and deposition. If true for areas beyond our study site, the observed association between cold and windy conditions has significant implications for the perceived sensitivity of regional shorelines to coastal erosion and ensuing societal impacts like infrastructure damage and carbon release[13,19,74]. At the same time, we would like to stress the compounding impacts of two processes that will affect Arctic coastal dynamics independent from changes in wind and wave energy: permafrost degradation and sea-level rise, which are mostly thermally driven[17]. Last but not least, it is important to bear in mind that our reconstructions capture summer variability, while recent modeling work suggests that future Arctic wind changes will be greatest during the fall and winter seasons[75].

## Wind extremes track ~1500-year internal climate cycle

As outlined above, SMI maxima coincide with Easterly and Westerly wind intensity extremes that occurred during regional cooling intervals as first suggested by Refs. 65,66, and shown in Fig. 5e. Both of these studies associate these recurring windy phases with quasi-periodic ~1500-year North Atlantic cold periods[76]. Wavelet transformation (see Statistics paragraph in our methods) reveals that this pervasive cycle also influences the spectral signature of both our PC-based wind reconstructions (Fig. S6). The Steinbruvatnet record adds to our understanding of this climate cycle. Notably, our PC 2-based reconstruction extends the temporal evolution of the spectral signal of the Westerly winds beyond the 6000-year perspective provided by Ref. 66. And while our data confirm the influence of a ~1500-year periodicity during this period, they also show that shorter cycles become more dominant further back in time (Fig. S6). This so-called Mid-Holocene transition has been detected in numerous proxy reconstructions from around the world[77], and is often associated with the concurrent stabilization of large-scale climate boundary conditions like sea-level and ice sheet extent[78]. But unlike these studies, we find no conclusive evidence that the ~1000-year cycle characteristic of solar forcing dominates during the Early Holocene (Fig. S6). However, regarding higher-frequency variability, our Westerly wind reconstruction does indicate an increase in the amplitude of decadal-scale variability during this period (Fig. 5). This observation contrasts with the results of Ref. 79, who infer dampened Westerlies variability during the Early Holocene based on changes in varve thickness, although we should note that this reconstruction records winter conditions, whereas our data records summer change (see the Holocene evolution of Steinbruvatnet). If true, our findings can be relevant for the predictability of regional wind change, as the future will be shaped by melting ice sheets and warming like the Early Holocene. However, as with the inferred association between cold and windy conditions (previous paragraph), additional reconstructions from the wider region are required to assess the representativeness of our data. Preferably along a spatial transect to trace latitudinal shifts in wind strength through time as also suggested by Ref. 64.

## Methods

### Coring

We analyse the 106 cm long sediment core 601-21-6 GC, which was collected from Lake Steinbruvatnet in August 2021 at a depth of 1.86 m using a Uwitec gravity corer (76°29′N, 16°33′E). We targeted a flat section in the central part of the lake to avoid disturbances (Fig. 1c). Following fieldwork, the core was split lengthwise, then visually logged and photographed with an RGB line camera that was attached to an ITRAX scanner (see below).

### Stratigraphy

Following logging, we first conducted several non-destructive scanning analyses. X-Ray Fluorescence (XRF) scanning was performed on an ITRAX scanner at the EARTHLAB facility of the University of Bergen (UiB) to map fluxes of eolian minerogenic elements and sea-spray aerosols[29]. To measure minerogenic input with higher atomic numbers with greater precision, the scanner was fitted with a Molybdenum (Mo) tube set to 40 kV and 10 mA. Down-core measurements were generated for 34 elements at 200 μm intervals. We selected XRF elements with **1**) high sensitivity to the fitted Mo tube, and **2**) a Signal-to-Noise ratio (SNR; $\mu/\sigma$) higher than 2 after[54,61]. XRF data are presented as **1**) Total Scatter Normalized (TSN) ratios for individual elements after[29,54], to account for variations in organic and water content[29], and **2**) log ratios when looking at proportions between two elements after[29,80]. Computed Tomography (CT) scanning was applied to visualise sediment structures like storm layers in 3-D[54,81], and to determine ensuing variations in density captured by CT grayscale values[53,54]. CT scanning was performed on a ProCon X-ray CT-ALPHA scanner, operated at

110 kV and 810 µA, with a 267 ms exposure time to generate ca. 100 µm resolution 24-bit scans. Scans were then processed with version 9 of the Thermo Fisher Avizo software to generate 2-D orthoslices and 3-D reconstructions. Subsequently, we used CT orthoslices to verify initial visual logging and create a schematic lithostratigraphy after[31] (Fig. 1). To this end, we relied on the image trace operator in Adobe Illustrator CC 2015[82]. Next, we performed destructive physical analyses to measure down-core variations in organic content, density, and grain size distribution. Based on CT imagery and visual assessment, we extracted 97 samples with a 0.3 cm wide 1 ml syringe from minerogenic (facies 2) layers ($n = 79$) as well as organic background (facies 1) sediment ($n = 18$) at irregular 0.2–3.5 cm intervals. All samples were dried 12 h at 105 °C and then combusted for 4 h at 550 °C to determine Dry Bulk Density (DBD; g/cm$^3$) and Loss on Ignition (LOI; %; a measure of organic content)[83,84]. Grain size, a commonly used indicator of wind strength[57], was measured on all 97 sediment samples from the core, as well as 4 catchment samples from active eolian deposits near the western (CS 1) and eastern (CS 2-4) lake shores (see Fig. 1c, d), using a Malvern Mastersizer 3000 with a Hydro SV dispersion unit. Each sample was measured 5 times to warrant reproducibility. Following the recommendations of Ref. [85], sample particle size distributions were processed using the GRADISTAT software and expressed as metric (µm) Folk and Ward measures. Finally, End-Member Modeling Analysis (EMMA) was applied to the core samples to unmix particle size distributions and their sediment sources[59]. The analysis was run with the AnalySize 9.3 tool in MATLAB[86]. We used the non-parametric HALS-NMF algorithm which is well-suited for improving the unmixing accuracy[87], and thus identifying End-Members (EMs) and their abundances[86].

## Chronology

We relied on radiocarbon ($^{14}$C) dating to establish age control. To allay concerns about freshwater reservoir effects, we primarily picked terrestrial plant fragments (leaves and stems). As can be seen in Table 2, the only exception is aquatic moss sample Poz-177582, which was taken to assess offsets between terrestrial and aquatic material, but could not be dated with precision due to its small size. The material was extracted by wet sieving through 250 and 125 µm meshes, before overnight drying at 50 °C. In total, 13 samples were taken from 601-21-6 GC at semi-regular intervals and submitted for Accelerator Mass Spectrometer (AMS) dating in the Poznań Radiocarbon Laboratory, Poland (Poz)[88], and the Tandem Laboratory at Uppsala University, Sweden (Ua; Table 2). The latter was chosen in certain cases because it allowed dating of our smallest (2.7-7.1 mg dry weight) samples.

## Statistics

XRF and CT output was resampled on a common 0.5 cm with the lower-resolution physical analyses to allow multivariate statistical analysis. For this purpose, we employed a 0.3 cm (15 point) Gaussian smoothing operator to account for the width of the syringe used to extract samples (see stratigraphy paragraph in methods), before resampling at 0.5 cm intervals using linear interpolation in version 4 of the PAST software[89]. We used the same program to calculate Spearman's rank correlation coefficients ($\rho$) and to cross-correlate the results of physical analyses with the resampled data. To explore shared gradients of change captured by our independently measured eolian indicators, we carried out a Principal Component Analysis (PCA) on selected proxy parameters, using version 5 of the CANOCO software[90]. The input was centered and standardized before analysis, following software recommendations. Following Ref. [28], we detrended PCA output to account for the fact that (Early) Holocene sea-level changes influenced the distance between the lake and the coast[37], and thus fluxes of eolian material (see Core chronology). To this end, we relied on the remove trend transformation in version 4 of PAST[89]. We also used this software to

1) smooth PC 1 and 2 scores using a 15 point moving average to account for the lower-resolution (0.3 cm vs. 200 µm) of physical analyses, 2) cross-correlate selected timeseries, and 3) perform continuous wavelet transform (CWT) analysis to detect spectral signatures[89]. For this purpose, we used a Morlet mother wavelet, following the recommendations of Refs. [78,89]. Also, considering the uncertainties of our chronology (see Fig. 2c), we only performed CWT analysis on the 3700–9700 cal. yrs B.P. interval bookended by radiocarbon ages. Finally, we applied WlCount—a semi-automatic lamination detection and counting software by Ref. [50]. By extracting visual information from the entire width of the CT image of investigated core 601-21-6 GC (see Fig. 3), this tool complements other down-core scanning data, which were acquired along specific down-core lines.

## Data availability

The authors declare that all data generated from sediment core 601-21-6 GC for this study, and presented in its figures and tables, have been made available in the DataverseNO repository, where the files can be accessed under the following https://doi.org/10.18710/TAUL5V. Source data are also provided with this paper as an .xls file with a tab for each figure. Source data are provided with this paper.

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

## Acknowledgements

This research has received funding from the Polish National Science Centre grant *ASPIRE - Arctic storm impacts recorded in beach-ridges and lake archives: scenarios for less icy future* No. UMO-2020/37/B/ST10/03074. Willem van der Bilt's contribution was supported by a Starting Grant (TMS2021STG01) from the Trond Mohn Research Foundation (TMF). We thank C.J. Hein, S. Lindhorst, M. Kasprzak, J. Kavan, M.F.A. Furze, E.W.N. Støren, A.G. Auer, and J. Buckby for their support during fieldwork. We thank the crews of S/Y *Ocean B* and S/Y *Pacific Star* for safe passage to Sørkapp, and Sysselmesteren to allow us to collect material there under permit 11680 in the Research in Svalbard (RiS) database. We also express our gratitude to R. Stange for sharing his aerial photographs of Sørkappøya. Finally, we thank Ł. Maciąg for his assistance with grain size analysis, T. Goslar, A.E. Bjune as well as T. Lin for their help with [14]C dating, M. Debret for sharing his recommendations on CWT analysis, and J. Karstens for his helpful comments on the manuscript prior to submission.

## Author contributions

W.v.d.B. and M.C.S. obtained funding this study. The study design and applied methodology was developed by W.v.d.B. and applied by Z.S.

Fieldwork and sediment data collection was carried out by W.v.d.B. and M.C.S. Z.S. and W.v.d.B. wrote the original draft. W.v.d.B. led the revision process, while Z.S. and M.C.S. also contributed to the final manuscript.

## Funding

## Competing interests
The authors declare no competing interests.
