## [Transparent Peer Review file · Nature Communications]

Coastal lake sediments from Arctic Svalbard suggest colder summers are stormier

Corresponding Author: Dr Willem van der Bilt

Version 0:

Reviewer comments:

Reviewer #1

(Remarks to the Author)

The submitted manuscript presents a multi-proxy record from a lake on Svalbard, which is presented as a reconstruction of storminess spanning the Holocene. While I have questions about some of the data, analysis and interpretations, detailed below, some of the results fill a gap showing past storminess in a remote and vulnerable region. The paper is novel because it fills a knowledge gap, as there is a lack of storminess reconstructions from high Arctic locations such as this. Aside from the issues listed below much of the data analysis has been done well, the paper is well written and figures presented clearly.

My main concern with this manuscript is the data analysis and interpretation of the ITRAX data, which unfortunately has been the focus in this manuscript. ITRAX data is a semi-quantitative measure of element concentrations, with the 'counts' of each element influenced by factors including the organic content (known as the dilution effect) and the concentration of other elements (the matrix effect) (e.g. Lowemark et al., 2011; Croudace et al., 2006). The core in this study has large variations in organic content, as shown by the LOI results and visible mineral rich layers, and this appears to have strongly influenced the element results (i.e. there are similar changes in Ti and Br/(inc/coh) to those of LOI). The PCA analysis shows that PC1 has loadings of ITRAX elements (Ti and Br) with high positive loadings and negative loadings of inc/coh. As the inc/coh is a measure of organic content through the core, it seems to me that PC1 really reflects the variations in organic content along the core, rather than easterly winds as is the interpretation.

The authors state that all the ITRAX data has been normalised using the inc/coh measure of organic content to avoid this issue (called total scatter normalised ratios), but has this method been robustly tested in the cited papers or elsewhere to confirm that it does remove the organic influence on the data? Perhaps normalising against a conservative element and a log-ratio transformation might be a better approach, as this has been tested (i.e. Weltje and Tjallingi, 2008; Orme et al., 2023). It would also be good to clarify in figure 3 if the Ti results have also been normalised using the inc/coh, as you appear to do with Br in this figure.

Given that I would argue the itrax results and PC1 reflect variations in organic vs mineral content through the core, this then makes me question some of the interpretations in the following discussion, particularly that PC1 and PC2 are reflecting easterly and westerly winds. For example at lines 342-344 there is a comparison between the PC1 record and the Jackson aeolian record of storminess from Iceland, with the conclusion that both represent easterly winds. A first point is that I cannot find where in the Jackson paper it says this record is caused by easterly winds. A second point is that if PC1 is reflecting organic/inorganic content, then the PC1 could be reflecting changes in temperature, which can influence sediment deposition in various ways. Temperature could impact on the duration of summer ice-free conditions when sand could be deposited in the lake; the duration of snow cover each year influencing aeolian entrainment of catchment material; the production of organic material within the lake ecosystem with a diluting effect on mineral material; the amount of moss and vegetation cover on surrounding landscapes influencing aeolian erosion; the sea level and availability of coastal sediments for erosion, etc... A more careful or balanced interpretation is required I feel.

While I don't trust the ITRAX data currently or the interpretation of Easterly and westerly winds, I feel that there is value in the other results. The visible layers, loss-on-ignition, grain size data, density and End Member analysis results support that the mineral rich layers deposited in the lake probably resulted from periods of high storminess. The ITRAX data supports this as well, if it is understood as a high resolution record of organic vs inorganic content. The PC2 record captures changes in Zr/K and the grayscale record, which are more reliable (as Zr/K is normalised and grayscale is identifying density changes along

the core), and therefore PC2 is likely reflecting changes in the mineral layers and storminess. This record shows similar results to the Jackson Iceland record and stacked record by Sorrell, so helps build a picture that these were stormier periods even in the high Arctic. Overall, the non-ITRAX records here do enhance our knowledge of past Arctic storminess, but not with the current analyses and interpretations.

Other points:

- Line 30 'reconstruction of arctic wind and wave strength' – you haven't provided a record of wave strength, it is aeolian activity. If overtopping is a factor then Holocene changes in sea level become much more important in this setting. 'wave blown sediment' at line 34 is also an unusual statement, does this refer to bromine?
- Lines 64-68: it states here that future changes in storminess in the Arctic are uncertain, which I would agree with. However elsewhere in the manuscript (line 36, line 98, lines 365-372) you say that there is a widely held view that the future Arctic will be stormier, which is contradictory.
- Line 105 – 'only modest sea-level changes after deglaciation ~11,000- 9,000 cal. yrs B.P. as shoreline uplift has not exceeded 10 m over the last 6,500 years' – what has been the relative sea level change, is it 10m? Or is this the isostatic change? I wonder if a 10m change in sea level could have altered the formation of the areas geomorphology (through changes in sediment supply) and whether this could have then altered the availability of material for aeolian transport into the lake.
- Line 110-112 (and Fig 1d) – would be good for the geomorphic features to be labelled on figure 1d
- Paragraph starting at 161 – the explanation of the radiocarbon outliers here is quite unclear. It could be possible that the very old ages dated to ~10,000 years old in the upper part of the core are from reworked land deposits as suggested, although if possible some additional dating controls would be beneficial to confirm the core top age. However I don't know what relevance the following explanation about sea level has to this interpretation. Furthermore, the paragraph in one part says that the sea was close to the level of the lake at ~8 ka BP (line 172) altering the sedimentation rate in the lake, but later (at line 180-182) says the evidence supports that the lake was isolated from ~10 ka BP and at lines 218-222 that there were no storm surges after 9.5 ka BP. If there was a marine influence on the lake up until 8 ka BP this would have implications for the lower section of the record and should be discussed.
- Table 1 – the LOI correlates most strongly with the inc/coh, bromine and Ti results, supporting my above point that organic content is an influence on the itrax data, even when normalised by the inc/coh data. The caption might be wrong: the italics in the table show insignificant results, not those greater than 0.05 probability. I think the 'greater than' sign is the wrong way around.
- Lines 278-280 – The point here that the strong correlation between PC1 scores and bromine supports that PC1 and bromine are both storm proxies may also be incorrect. As PC1 and the ITRAX results are potentially reflecting organic changes along the core, the bromine ITRAX results are also influenced by this issue. Furthermore bromine can be bound to organic material, therefore an increase in organic material would lead to more bromine in the sediment. This correlation with Br is also then used to suggest the results reflect the summer season when the adjacent sea is ice-free (line 281), which may not be the case if the results are mis-interpreted.
- Line 288-290 – sea spray (bromine) is suggested as not being deposited when winds are too strong, as a way to explain the lack of correlation between bromine and PC2. Evidence does not support this hypothesis because sea spray deposition increases exponentially with wind strength (e.g. Franzén, 1990; Gustafsson and Franzén, 1996; Meira et al., 2008). Instead I would say the lack of correlation is because bromine and PC1 reflect organic changes, and PC2 reflects grainsize and sediment density changes.
- Line 305 – 'reflect windy events or phases of stronger winds' – not sure what the difference is? Are windy events single severe storms?
- Line 322 – the decision to choose just 2 storm records/papers is narrow, and suggests these have been chosen because of their similarity with the record presented. Papers such as Goslin 2018 extend back to the early Holocene, while the Sorrell paper only goes to 6.5 ka BP, like other more recent records (e.g. Kylander et al., 2019). A more comprehensive comparison could go in the supplementary information.
- Line 331 – the storm magnitude index does not add much to the paper, as the clusters of dots in Figure 5 simply align with the peaks observable in the PC records of this figure.
- Line 353-355 – the explanation for differences between PC2 and the Sorrel storm stack record doesn't make sense, because the storm stack in Sorrel is not just based on foreshore archives, but includes sand dune reconstructions and peatland storm records too.
- Figure 5 – there does seem to be similar timings between the PC2 record, the Jackson peaks in aeolian activity and the Sorrel stacked record, suggesting that there are widespread increases in storminess, which is an interesting finding
- Paragraph from 361 – here 5 stormy intervals are defined, so perhaps it wasn't necessary to distinguish between easterly and westerly winds? Later in the paragraph the interpretation changes back to E and W storm records again. Given concerns

about PC1, I would just define storm events in the record and discuss these, or use the end member results to show changes in E and W winds.

- Lines 396-399 (and figure S7) – the original Fe record by Darby in figure S7c is not shown (also the IRD and MSG original records), so it is difficult to determine whether there is a convincing similarity in the variability of the Arctic Oscillation reconstruction and PC1 record. The conclusion that a 500 year lag with the 1500-year cycle in the Holocene means there was a different forcing is not necessarily correct, could it perhaps be due to chronological uncertainties between different records, or feedbacks in the climate system?

- The results of the spectral analysis again separate easterly and westerly winds (PC1 and PC2), which as covered above may be an issue if PC1 reflects organic variability and other drivers, such as temperature perhaps.

Reviewer #2

(Remarks to the Author)

This is a very interesting paper. The rationale behind the paper is well described and reasoned. In particular, the data collected, and the statistics used have been applied in a well-thought-out manner. The resulting coherence between phases of increase wind strength with existing records from the region is a remarkable and novel finding, as is the fact that stronger Easterly winds primarily occur during colder periods. The authors argue that their data provides a link between the stronger Easterly winds and the Arctic Oscillation, but also that the Westerly winds have a different casual mechanism - even though both have a 1500-year cycle. The application of the Storm Magnitude Index (SMI) to quantify the strength and intensity of storm is an especially interesting concept.

The methods and data analysis are generally sound. To reliably draw the conclusions the authors have in this paper, however, there needs to be more age control. The main concern is that there are no reliable dates in the top 20 cm, covering ~4000 out of ~10,000 years of this lake record. In terms of time, nearly 40% of this record is essentially undated. This matters because the last 4000 years covers nearly two out of six (or $\frac{1}{3}$) of the 1500-year cycles that could be present in this record. To make conclusions about cyclicity of this timescale and the implications of stronger winds in general, and the Easterly SMI during the late Holocene (as shown in Figure 5a) in particular, there needs to be a reliable age constraint every 500-1000 years throughout in the record. A Pb-210 dated modern section near the top of the record with wind and storminess proxies compared to observational meteorological data would provide greater confidence in the interpretations and conclusions.

There are some missing words and other minor typos in the text which are highlighted in the marked-up PDF.

Comments and question from the PDF are reproduced below:

Line 167: Why did you not also use Pb-210 and/or Cs-137 dating to establish that the last 100 years had been captured at the top of the core? There is currently no age constraint for the last 4000 years in this record so all age data in this part of the record can only be considered, at best, estimates.

Line 177: There is only one example reference included here (probably for reference limits). Nevertheless, are there any other examples because you say that it is often used. Ca has many potential sources. Is the geology as well as the setting of this example similar to your study site?

Line 178: This phrasing is confusing as it implies a double negative - the inclusion of the brackets is unclear

Line 216: Can you say what the other sources of Br are in your record and how you assessed the potential for Br mobility? How could Br mobility affect your interpretations when used as a sea spray proxy?

Line 309: It is good to highlight this as a potential drawback, but it leaves the reader wondering why a calibration with observational data wasn't attempted at this site. Did you collect and date sediments that overlap with the modern observational era from this site?

Line 393: This figure is important and should be included in the main text

Figure 3: Is Ti cps or % TSN or % cps sum? The caption says counts but these would be whole numbers if that was the case. It would be useful to also show the measured Br data, in the same way that you have plotted the Ti data. For single element plots it might be worth considering plotting these as centred log ratios (clr) to account for the closed sum effect of XRF scan data - see Bertrand et al. (2024) for further reasons.

Line 483: Why did you not also use Pb-210 and/or Cs-137 dating to establish that the last 100 years had been captured at the top of the core? There is currently no age constraint for the last 4000 years in this record so all age data in this part of the record can only be considered, at best, estimates.

Line 507: All of these methods would OK to apply across the whole record if the top 4000 years of the record had age constraint - as it stands, it would only seem suitable to apply these methods for the 4000-10000 year BP section which has some age constraint.

Why did you choose 150-year moving average? Does this bias the output of the wavelet analysis in any way?

The supplementary material is excellent and the figures in enhance the paper substantially. It would be nice to make a summary figure for inclusion along with Figure 2 in the main text showing, for example, one of the CT scan outputs where there is room to do this. Figure S6, the wavelet diagrams, would add great value to the main paper too.

References:

Bertrand, Sebastien, Rik Tjallingii, Malin E. Kylander, Bruno Wilhelm, Stephen J. Roberts, Fabien Arnaud, Erik Brown, and Richard Bindler. 'Inorganic Geochemistry of Lake Sediments: A Review of Analytical Techniques and Guidelines for Data Interpretation'. *Earth-Science Reviews* 249 (1 February 2024): 104639. <https://doi.org/10.1016/j.earscirev.2023.104639>.

Stephen Roberts

Reviewer #3

(Remarks to the Author)

Reviewers report to "Not gone with the wind: 9,500-year sediment record of Arctic storminess favors internal climate control"

This manuscript presents an interesting data-set from a region that is currently undergoing rapid change, and where limited previous research into paleo-storminess variability have been done. The authors have applied a multiproxy approach and analysed a sediment core from a small lake located at beach in the southern-most tip of Svalbard. The sediment sequence was analysed for grain size (Malvern 3000), chemical composition (CS-XRF) and visual variability by computed tomography. Selected proxies and their relationship were assessed through principal component analysis (PCA). The authors thereafter used principal component 1, supported by a ratio between Br and Inc/Coh ($Br/(Inc/Coh)$) as a proxy for changes in easterly winds. Principal component 2 are used as a proxy for westerly winds. Most of the figures are clearly and nicely organised.

Although the study represents a highly relevant topic and interesting data-set, it needs considerable re-working prior to publication. Some of the issues that needs to be resolved include:

1) The chronological control

The temporal uncertainties are considerable in this record, which needs to be better recognised and taken into consideration. For example, 4 out of 9 ^{14}C dated samples showed reversed ages. After 3.8 ka (above 29 cm) all ages are reversed, meaning that there is limited (no?) chronological control after this period. The age-depth model was done using Clam, which does not depict age-uncertainty in a good way. I would recommend updating the age-depth model using R.Bacon. See enclosed example of how an updated age-model for this record would look.

2) Transport processes

The site is located only 5 masl, but still only aeolian transportation is assumed. The motivation for this interpretation is not clear. Larger clasts and pebbles are also found in the deposit. The authors interpret these to be drop-stones deposited by melting ice during spring melt. If this is correct, other minerogenic material may also be deposited during the spring melt, and not via wind transportation.

3) Data handling

The processing of the CS-XRF was not conducted in accordance with the latest recommendations. See eg. Bertrand et al. 2024 (<https://doi.org/10.1016/j.earscirev.2023.104639>) for a proposed, and updated, protocol.

Also, a ratio between $Br/(Inc/Coh)$ is used as support for interpreting variability in wind easterly wind strength. However, concentration variability of Br have been shown in multiple papers to be related to organic matter and decomposition processes, so this ratio likely does not represent changes in grain size or mineralogy which needs to be revised.

See e.g. Martínez Cortizas et al 2016 (<http://dx.doi.org/10.1016/j.gca.2015.11.013>).

The Inc/coh ratio may reflect organic matter content, but also varies with other properties, such as water content which was not taken into consideration (see Bertrand et al. 2024).

4) Interpretations

The interpretation that CP1 represents changes in easterly winds while CP2 represents westerly winds are presented in the manuscript. Unfortunately, the assumptions that these inferences are made on are highly uncertain. For example, CP1 is argued to represent easterly winds as one of the variables included in the PCA (EM1) is supposedly similar to the source sample collected immediately east of the lake, while CP1 (EM3) is interpreted as similar to the source sample collected immediately west of the lake. The results reported in (S. Fig 4) do not support this. EM1 and EM3 both show considerable overlap with the eastern beach sample (CS1), and no end member matches the western beach which is much coarser (450 μm) than EM3 (~80 μm). In addition, the beach sediments are likely a major sediment source of minerogenics deposited in the small lake, but no sample from the beaches were retrieved, instead two particular sediments were sampled 1) a silty sheet deposit (immediately east of the lake) and 2) a shadow dune (immediately west of the lake, below a bedrock outcrop). This means that the grain size composition of the most common sediment type in the area remains unknown, and thus, any west or east differences in composition are also unknown. In conclusion, it is highly unlikely that these results can be used to infer changes in wind strength in the easterly and westerly wind patterns.

Due to the arguments presented above I do not think that the conclusions presented in the manuscript, in its current state, can reliably be made. However, I do think that this data-set can become an important contribution to the field, following re-

working and re-interpretation of the data.

I would suggest re-processing the CS-XRF data, updating the age-depth model, acknowledging the age-uncertainties and taking a closer look to the grain size results from this record. Periods of coarsening grain size distributions may be used to infer periods of strong winds and storminess, and maybe even to infer main transport processes (see suggestions in the supplementary file).

Version 1:

Reviewer comments:

Reviewer #2

(Remarks to the Author)

I have read the reviewers revised article and their response to reviewers report, which is very thorough and comprehensive. The authors have undertaken a substantial revision to the previous submission and have addressed all previous concerns in a sensible and realistic manner.

Overall, their more cautious interpretation of the results in the paper is admirable, and, in my opinion, does not detract in any way from either the significance or novelty of their findings. In particular, I am pleased to see that they now address the shortcomings in the record's chronology and the impact that this has on establishing cyclicity in parts of their record that are not well-dated, i.e., the last 3,000 years. Their approach to dealing with Br also makes sense, as this element is notoriously difficult to interpret in terms sea spray from ITRAX data alone, especially where large shifts in organic content exist downcore.

One additional (minor) comment on the revised submission, which they may wish to consider, relates to the use of %TSN for Ti and how the authors dealt with noisy clr transformation. Although some previous papers use the %TSN method (and the guidance from the Cox analytical is it remains a valid normalisation technique for taking variations water and organic content in scan data into account), it is probably worth noting that %TSN produces the same result as calculating the % of cps sum. The latter is mathematically more straightforward and understandable, mainly because it is analogous to transformation of species count data to percentage datasets. The %cps sum is also described as a possible data transformation method in our recent Bertrand et al. (2024) paper. My personal preference these days, following final recommendations in the Bertrand et al. (2024), would be use clr transformed datasets when dealing with individual element profiles. As the authors discuss in their response document, there are some drawbacks with the clr method, mainly that it produces very noisy element profiles when some elements contain many zero cps datapoints. Clr and log ratios both involve the use of log transformation of elements with zero values, which is especially problematic for clr as multiple elements are incorporated into the calculation and because the whole row is then lost if any of the elements contain zeroes. However, it is possible, and statistically valid, to replace zero values with either half each elements' minimum value or by a value that is a randomly generated fraction of the detection limit for each element prior to clr (or log ratio) transformation. While zero replacement should be considered carefully for each individual dataset, zero replacement will enable the retention as much non-noisy element data as possible (i.e., elements without many zeros). The clr process and zero replacement can be undertaken rapidly within R and/or using the 'zeroreplace' function in the 'Compositions' R package - it would be a good idea to read the caveats in the package notes before proceeding with this. Bertrand et al (2024) also recommend using incoherence normalised log ratios when investigating individual elements, in this case $\ln(\text{Ti}/\text{inc})$, for reasons outlined in the paper - this approach would be more consistent with the reasoning the authors use elsewhere in the revised submission wrt Zr.

Reviewer #4

(Remarks to the Author)

The authors present lake core data from the southern tip of Svalbard, Sørkappøya island, from which they conclude that "a colder Arctic is stormier." The island has a unique location in the sense that it appears to preserve an interesting record of aeolian deposition during summer months. The main conclusion of the paper is that "we reveal quasi-cyclic wind maxima during regional cold periods "... (which) challenge the emerging view that a warmer and less icy future Arctic will be stormier." Unfortunately, the main conclusions are not supported by the data, due to the following three points.

1) As mentioned by the authors, the lake is covered by ice during 9 months, and is therefore only receptive to direct aeolian deposition during a few summer months. However, storms are most intense and frequent during winter. Therefore, the data can not be used to assess general "storminess", only storminess during summer, and the main conclusion is thus not supported. In the discussion, this seems to be overlooked and the results are compared to annual storminess from other records (see the Iceland and Outer Hebrides studies).

- a. Why is the relationship between summer and winter storminess not even discussed?
- b. How did lake ice coverage change throughout the Holocene, and how does this influence the storminess reconstruction?
- c. In line 404, the authors state that "our data is biased towards summer", but really the data only relates to summer storminess.

2) Presenting data from one site in the Arctic is not enough to make conclusions about reconstructing "quasi-cyclic wind maxima during regional cold periods" and to evaluate whether "a warmer and less icy future Arctic will be stormier".

Throughout the manuscript it is implied that the data is representative for 'the Arctic' (what region do the authors mean by this? The whole Arctic? Svalbard?). To support the claim, the community will really want to see how reproducible these results are, for example across Svalbard, before a claim regarding the Arctic can be accepted. In NW Europe, it has proven difficult to reconstruct general patterns of overall storminess, simply because climate change can cause storm tracks to shift. Would this be different in the Arctic? Any discussion on the representativeness of the site for 'the Arctic' is entirely lacking, as well as a discussion on potential storm track shifts, presenting a major flaw.

3) The authors have solid and interesting primary data (grain-size analysis, end member modeling, XRF data), but they heavily base their interpretation on a derivative of the primary data, relying on PCA analysis. Few arguments are presented why the PCs are interpreted the way they are, and I am concerned about the trustworthiness of the PCA analysis due to the following reasons:

- a. The variability of the PC2 looks spuriously similar to PC1. This is odd because PC2 is supposed to capture the 'leftover' variability that is not captured by PC1. I quickly checked the data and the Pearson correlation coefficient between PC1 and PC2 is 0.55 — this is way too high and likely points to artifacts created by the data analysis/preparation. I recommend the authors to thoroughly check their PCA analysis.
- b. In the PCA biplot no clustering of the sample scores is visible, they all sort of plot around the origin. Therefore it is quite unconvincing that the PCA is really capturing underlying structures of the data. It is OK to try PCA on a dataset to see if patterns can be discerned, but one should critically evaluate the results. It should also be noted that PC2 only captures 11% of the variation.
- c. The most direct proxy for wind strength, i.e. actual physical grain-size measurements, are not taken into account in the PCA analysis. It would be much better if the authors make their case based on the primary data, rather than an unconvincing PCA analysis. As a result, the PCA-derived SMI index is also strongly jeopardised because of this.

4) I share the concern with reviewer 3 that if cobbles and pebbles can be ice-rafted, then why not some of the fine sand? The authors know the study area best, but reading the response letter it is still not clear to me whether (fine) sand is presented in the vicinity of the lake that could be rafted to the basin. Are the authors willing to state that "there is no fine sand present around the lake that could be transported by lake ice into the lake" (if it is true)?

Other comments

- A map of modern storm tracks is really missing.
- Please provide a transition clays/silt/sand diagram in Fig. 2/3. Please provide a litholog and a plot showing mean grain size changes.
- Captions of the Supplementary figs are missing?

Reviewer #5

(Remarks to the Author)

The ms, based on a lake core from a strategic location at the southern tip of Stitsbergen, gives a unique multi-millennia scale record of Holocene changing wind-systems over the Arctic and North Atlantic — a key area in the debate of Global change. As such, it will contribute to the rapidly growing literature on the importance of storminess in future climate change. Contrary to some earlier literature, the record indicates that future warming should not increase storminess in these parts.

The c. 1 m lake core is made up of eolian silt, blown in from both east and west, and treated with an array of sophisticated instrumental and statistical methods to distinguish source areas and wind intensity. I am not an expert in these fields and will not comment on them, although they are a very important part of the work. Hopefully others will take this up. (The authors appear to me as competent and with a sound critical approach to their results). All this evidence is interpreted in terms of the local coastal morphology and sedimentation.

In the discussion the results are compared to other multi-millennial scale records from coastal NW Europe, finding a broad agreement with these records with cold periods being stormy, although there is some disagreement about the wind situation during Holocene warm period (see f.i. the recent paper by Sjöström et al., QSR 2024).

In general I find the ms well written with relevant and well argued points, but I do have a few suggestions for the authors to consider.

The role of lake ice is referred to only sporadically, but deserves a closer look. Presently the lake is ice-free for only three months of the year (which may explain why only c. 1 m sediment has accumulated over c. 10,000 years). Therefore, the record is strictly a summer record, leaving out the winter, which is usually the time for storminess. This should be considered, not least when comparing to lake records from more southerly areas where lake ice is absent. Also, the annual duration of lake ice must have varied over the 10,000 years. An increase or decrease of one month per year, which is not an unreasonable range, would prolong or diminish the window available for sedimentation with 25%, with consequence for sedimentation rates, independent of such factors as proximity to the coast.

I suggest that the local RSL curve, which is an important parameter and frequently referred to, is reproduced together with the age model in Fig. 2.

The age model, based on 13 C14 dates, is marred by "contamination" from a discrete, but unknown source of old remains of terrestrial (why?) plants. Eight dates from throughout the core apparently represent this material only, while five are considered to be "clean" with no contamination. This does shed some doubt on the model, and it would have been useful with some OSL and/or magnetic declination dates, but the authors give a fair discussion of the problems, and the model

stands with its uncertainties as best solution to the problem.

In summary, the core gives important new evidence on Holocene storminess from a unique locality in the Arctic. It is well researched with state of the art methods and a comprehensive survey of relevant literature from the North Atlantic coasts, claiming that the future along presently sea ice covered coasts may not be as stormy as anticipated by others. It should be of interest to a broad audience of climate researchers.

Version 2:

Reviewer comments:

Reviewer #4

(Remarks to the Author)

I have read the rebuttal and checked the revised version of the manuscript. Overall, the authors have presented a much more honest representation of what the data does and does not show. The study still has weaknesses but can be published. Below my evaluation of the concerns previously raised.

1) Previously, the study was presented as if the data could be used to reconstruct Arctic storminess. The authors now acknowledge that, actually, the study presents a record of summer windiness, during the time of the year when few storms occur. They acknowledge that the provided data may or may not be representative of changes in the broader region. They have made this very clear throughout manuscript, and adjusted their title. I appreciate these changes, as they have markedly improved the manuscript.

2) With regards to site-specificity and extrapolation of the the results, the authors have now toned down their claims and acknowledge that the results may, or may not, represent broader changes in the region and that further research is required.

3) My third comment was that more evidence should be provided to support the interpretation of the principal components, which the study heavily relies on. The authors are very confident in their interpretation of PC2, which is predominantly loaded by the CT values, yet there is no concrete evidence that links the CT values (or PC2 as a whole) to the westerlies. A convincing way in which the PCA analysis could have been supported would be by showing a comparison of the PC scores with modern/historical wind data. However, due to the disturbed core top the authors were not able to do this. Therefore, this analysis and interpretation remains a significant weakness of the study, but I suggest the study is published and the community can form its opinion. The authors have also showed that the the PCs are independent – this was a mistake on my part and my apologies for that. The authors did well by now converting their dataset into a standardized data format. They also point out that PCA can be used to tease out “hidden” signals in the data. Of course this is true –and a PC that captures 11% may or may not be ‘significant’ – but my point was that with every PC the risk of picking up noise does increase. Therefore, additional evidence should be provided that a certain PC represents a certain physical phenomenon, i.e. that it is meaningful. Again, this is very important in the case of this study, since no comparison of the PCs with modern/observational data is provided.

4) The concern of ice-rafting is resolved as the authors clarify that only pebbles are available for ice-rafting.

Reviewer #5

(Remarks to the Author)

Thanks for the revised ms and the very comprehensive discussions. I was glad to see that my main point - stressing that this is a record of summer storminess only - has been acted on, both in the title and in the text. I was surprised to see that another of my points - the role of varying snow and lake ice cover for sediment input into the lake - apparently has had little or no impact, but I am satisfied that this issue has now been addressed.

From reading the other reviewers' comments from fields that I am not really into, my feeling is that this ms is now ready for publication - I agree.

Revision – manuscript NCOMMS-23-56354

We express our sincere gratitude to the reviewers for their valuable time and insightful comments, which have significantly improved our manuscript. **1)** revised our chronology by incorporating additional radiocarbon dates and re-running the model in Bacon to reflect increased uncertainty margins (reviewers 2 and 3), **2)** re-analyzed our XRF data by following the recent recommendations of Bertrand et al. (2024) and providing a more detailed explanation of our normalization approach (reviewers 1 and 3), **3)** limited our analysis of cyclicity to more accurately address the limitations of our chronology (reviewers 2 and 3), and **4)** enhanced the connections between catchment geomorphology and lake sedimentology, mostly by including more catchment samples, to better support process-based interpretations (reviewers 1 and 3). We have carefully addressed all reviewer comments point-by-point in the text below, with our responses highlighted in *italics*. All additions and changes to the manuscript are tracked.

Reviewer #1 (Remarks to the Author):

The submitted manuscript presents a multi-proxy record from a lake on Svalbard, which is presented as a reconstruction of storminess spanning the Holocene. While I have questions about some of the data, analysis, and interpretations, detailed below, some of the results fill a gap showing past storminess in a remote and vulnerable region. The paper is novel because it fills a knowledge gap, as there is a lack of storminess reconstructions from high Arctic locations such as this. Aside from the issues listed below much of the data analysis has been done well, the paper is well written and figures presented clearly.

We thank reviewer 1 for this encouraging assessment, and for the opportunity to revise.

My main concern with this manuscript is the data analysis and interpretation of the ITRAX data, which unfortunately has been the focus in this manuscript. ITRAX data is a semi-quantitative measure of element concentrations, with the ‘counts’ of each element influenced by factors including the organic content (known as the dilution effect) and the concentration of other elements (the matrix effect) (e.g. Lowemark et al., 2011; Croudace et al., 2006). The core in this study has large variations in organic content, as shown by the LOI results and visible mineral rich layers, and this appears to have strongly influenced the element results (i.e. there are similar changes in Ti and Br/(inc/coh) to those of LOI). The PCA analysis shows that PC1 has loadings of ITRAX elements (Ti and Br) with high positive loadings and negative loadings of inc/coh. As the inc/coh is a measure of organic content through the core, it seems to me that

PC1 really reflects the variations in organic content along the core, rather than easterly winds as is the interpretation.

The authors state that all the ITRAX data has been normalised using the inc/coh measure of organic content to avoid this issue (called total scatter normalised ratios), but has this method been robustly tested in the cited papers or elsewhere to confirm that it does remove the organic influence on the data? Perhaps normalising against a conservative element and a log-ratio transformation might be a better approach, as this has been tested (i.e. Weltje and Tjallingi, 2008; Orme et al., 2023). It would also be good to clarify in figure 3 if the Ti results have also been normalised using the inc/coh, as you appear to do with Br in this figure.

We used the TSN approach after the Nature Geoscience storminess publication by Saunders et al. (2018), at the time of writing, the benchmark in the field. As the authors outline in their supplementary, this approach was explicitly chosen to minimize the influence of organic content on XRF signals in their organic-rich lake sediment core. They, in turn, adopted this approach from Roberts et al. (2017) in Nature Communications, who applied TSN to deal with similar issues in a penguin guano deposit (also organic). We would, in this context, like to stress that TSN does not involve normalization against inc./coh., but entails subdividing the sum of an element by (inc.+coh.) +1. Regarding any (lingering) influence of organic content on our XRF data, we would like to stress that 1) each element (also the Ti) data shown in Fig. 3) has been TSN transformed (so – normalized against a measure of organic content that is substantiated for our study site using LOI as shown in Table 1), and 2) Br has not been included in our PCA (so has no loading on either PC – this, because it did not meet the Signal-to-Noise (SNR) criteria outlined in stratigraphy paragraph in our methods section. We apologize for confusing reviewer 1 regarding the various data analysis steps that were carried out and tried clarifying them in our revised manuscript. Finally, re-visiting the literature on this very important subject (XRF data processing), took note of the recent review paper by Bertrand et al. 2024 in Earth Science Reviews, which was only published after first submission, but holds important insights regarding XRF data processing. Following their, and that of reviewer 1, recommendation, we now apply log-ratio transformation to ratios used in our study.

We agree with reviewer 1 that there are, indeed, 1) large variations in organic content throughout the core (shown by both the LOI and inc./coh. results) and 2) visible mineral-rich layers, reflected by lighter-colored clastic horizons (facies 2) in the Computed Tomography (CT) picture (see Figs. 2 and 3). Unfortunately, we struggle to address the suggestion of the organic matter influence on the element results, as both Ti and Br (normalized by inc./coh.)

prove negatively correlated with both organic content proxies. We also want to stress that neither Br nor its ratios were used in the PCA. Finally, we would like to highlight that organic indicator inc./coh. has negative loadings on PC 1, and therefore we cannot support reviewer 1's suggestion that PC 1 reflects the variations in organic content throughout the core.

Given that I would argue the itrax results and PC1 reflect variations in organic vs mineral content through the core, this then makes me question some of the interpretations in the following discussion, particularly that PC1 and PC2 are reflecting easterly and westerly winds. For example at lines 342-344 there is a comparison between the PC1 record and the Jackson aeolian record of storminess from Iceland, with the conclusion that both represent easterly winds. A first point is that I cannot find where in the Jackson paper it says this record is caused by easterly winds.

Also, following the above, yes, the XRF Itrax data does track variations in minerogenic input (diluting in-lake organic production). Still, considering the site of our basin (coastal) and the accompanying grain size work (linking the granulometry of minerogenic sediment input to the lake to eolian sediment sources in the catchment), we argue that wind is the dominant source of minerogenic material to investigated Lake Steinbruvatnet (also see "The Holocene evolution of Steinbruvatnet" in the manuscript). We would also like to underline that this approach (linking minerogenic input measured, i.e., by XRF, but in this case also by grain size and CT, to eolian processes) is the general approach taken by (lake) sediment-based storminess studies (see, i.e., the excellent review by Kylander et al. 2023, in QSR). Regarding the interpretation of our PCs considering the Jackson et al. 2005 data, reviewer 1 is right that it is not easily distilled from the paper but follows from the analyzed Keflavik wind data (the revealed dominance of ESE and E winds) and how the authors link these to northeasterly winds in the results and discussion.

A second point is that if PC1 is reflecting organic/inorganic content, then the PC1 could be reflecting changes in temperature, which can influence sediment deposition in various ways. Temperature could impact on the duration of summer ice-free conditions when sand could be deposited in the lake; the duration of snow cover each year influencing aeolian entrainment of catchment material; the production of organic material within the lake ecosystem with a diluting effect on mineral material; the amount of moss and vegetation cover on surrounding landscapes influencing aeolian erosion; the sea level and availability of coastal sediments for erosion, etc... A more careful or balanced interpretation is required I feel.

While we hope that our previous explanation of our XRF data processing approach (explicitly focused on minimizing the imprint of diluting changes in organic content) resonate regarding the points raised here about organic productivity, we would like to stress that sea-level change 1) has been a critical factor in site selection (see i.e. Fig. 1b – showing the proximity of the 0 m emergence isobase to our study site), 2) is also discussed (and contextualized) in light of modelled sedimentation rates and other local sea-level evidence from beach ridges (see our Core chronology), and 3) accounted for by detrending our PC data to warrant assessment of changes against a stable baseline (see the statistics paragraph in our methods section). Regarding the possibly imprint of temperature changes on our PC data, we cannot fully allay these in the absence of an independent temperature reconstruction from the basin. However, we do present (and discuss) other evidence that at least does not favour a scenario where these have a dominant imprint on our data. Notably, the lack of coherence between local sea ice and (summer) temperature reconstructions and our storminess indicators (see Fig. 5). However, to accommodate reviewer 1, and be more realistic about the limitations of our work, we now stress that changes in snow, and ice-cover might have an impact on how minerogenic particles might be mobilized and deposited (see “The Holocene evolution of Steinbruvatnet”).

While I don't trust the ITRAX data currently or the interpretation of Easterly and westerly winds, I feel that there is value in the other results. The visible layers, loss-on-ignition, grain size data, density, and End Member analysis results support that the mineral rich layers deposited in the lake probably resulted from periods of high storminess. The ITRAX data supports this as well, if it is understood as a high-resolution record of organic vs inorganic content. The PC2 record captures changes in Zr/K and the grayscale record, which are more reliable (as Zr/K is normalised and grayscale is identifying density changes along the core), and therefore PC2 is likely reflecting changes in the mineral layers and storminess. This record shows similar results to the Jackson Iceland record and stacked record by Sorrell, so helps build a picture that these were stormier periods even in the high Arctic. Overall, the non-ITRAX records here do enhance our knowledge of past Arctic storminess, but not with the current analyses and interpretations.

We thank reviewer 1 for suggesting there is merit in our findings. Complementing the information in the above responses, we would like to stress that PC 2 not merely reflects bulk minerogenic input (facies 2 layers) like PC 1, but specifically denser and coarser minerogenic input (which is also supported by our grain size analyses and modelling). Based on this difference (and how it links to eolian sediment availability in the catchment – notably, the

predominance of silt-dominated material to the East, and exclusive presence of coarser sand-dominated material to the West: see Suppl. Fig. S3), we link our PCs to two different wind systems (see our “The Holocene evolution of Steinbruvatnet”).

Other points:

- Line 30 ‘reconstruction of arctic wind and wave strength’ – you haven’t provided a record of wave strength, it is aeolian activity. If overtopping is a factor, then Holocene changes in sea level become much more important in this setting. ‘wave blown sediment’ at line 34 is also an unusual statement, does this refer to bromine?

Reviewer 1 is right – we provide a record of wind strength (eolian activity) and have consistently changed this throughout the manuscript. Considering (other) reviewer comments regarding the inclusion of Br as a sea-spray indicator – which merely serves to add additional confidence to our PC 1 record – we have excluded this element from this revision.

- Lines 64-68: it states here that future changes in storminess in the Arctic are uncertain, which I would agree with. However elsewhere in the manuscript (line 36, line 98, lines 365-372) you say that there is a widely held view that the future Arctic will be stormier, which is contradictory.

We thank reviewer 1 for these suggestions. While we regret the confusion and have sought to remedy this (see below), we would like to stress that both notions contradict each other: there is indeed broad consensus that the future Arctic will be stormier, but it is based on highly uncertain information. We tried clarifying in the following ways:

Line 35-36: “Our reconstructions reveal quasi-cyclic wind maxima during regional cold periods, and therefore challenge the emerging view that a warmer and less icy future Arctic will be stormier”.

Lines 97-98: “Our findings suggest that Holocene wind maxima occurred during cold periods, and thus challenge the emerging notion that a warmer and less icy future Arctic will be stormier.

Lines 365-372: “By extending this association between cold and windy conditions into a study area that is seasonally sea ice-covered, our findings challenge the emerging view that a warmer and less icy Arctic will become stormier – the premise of this study (see introduction). This notion is supported by local evidence, which reveals that 1) Easterly winds were most intense

during the Late Holocene when sea surface temperatures were relatively low and severe sea ice conditions persisted in up-wind Barents Sea (Berben et al., 2017; Marchal et al., 2002; Risebrobakken et al., 2011), while 2) Westerly wind strength does not exhibit a clear relation with either temperature or sea ice conditions as SMI maxima occur throughout the Holocene (Figs. 1 and 5) (Müller et al., 2012)”.

- Line 105 – ‘only modest sea-level changes after deglaciation ~11,000- 9,000 cal. yrs B.P. as shoreline uplift has not exceeded 10 m over the last 6,500 years’ – what has been the relative sea level change, is it 10m? Or is this the isostatic change? I wonder if a 10m change in sea level could have altered the formation of the area’s geomorphology (through changes in sediment supply) and whether this could have then altered the availability of material for aeolian transport into the lake.

The max. 10 m refers to relative sea-level change and is largely based on the findings by Forman et al. 2004, in QSR. We should note here that, based on (yet) unpublished radiocarbon-dated whale bones from Sørkappøya, this number is likely closer to 5 meters. The absence of storm surge deposits in our sediment record, as well as the presence of undisturbed periglacial features (sorted circles) and gentle eolian features, all suggest the island has not been overrun by waves, even when the sea-level was a little higher than today. We hypothesize that this has to do with the geological and geomorphological characteristics of the island (also see Fig. 1d) – during fieldwork, we have not detected any overwash features along the coast. This supports our view that an underwater plateau dotted with rocky skerries, spurs, ridges, and platforms over an area of at least 1.0 km from the island's shores serves as wave breakers. This explains why, despite such a low position above sea-level, our lake has not been subjected to direct wave impact and storm floodings. The undisturbed development of the mentioned periglacial and eolian (micro)forms is another evidence of the absence of storm overwash and remodelling of uplifted surfaces. All this paints a picture of a system that, despite its location in one of the most storm-exposed areas in the whole of Archipelago, is resistant to direct storm impacts, with their effects limited to the influence of winds associated with the passing of cyclones. Finally, while comparatively modest, the inferred changes in post-glacial sea-level likely impact sediment supply, as reviewer 1 suggests. We also acknowledge this in our manuscript, notably by linking the inferred changes in sedimentation rates to local evidence of sea-level change, but also by detrending PC output to warrant an assessment of storminess against a stable baseline.

- Line 110-112 (and Fig 1d) – would be good for the geomorphic features to be labelled on figure 1d

We agree and have added a modified panel 1d with geomorphological detail for context. This new panel presents the diversity of geomorphological features covering eastern and western coasts, with key coastal morphodynamics controls – rocky ridges, spurs and skerries, which surround the island and break storm waves. We have also included sampling sites where we collected catchment samples (CS 1-CS 4) for granulometric analyses.

- Paragraph starting at 161 – the explanation of the radiocarbon outliers here is quite unclear. It could be possible that the very old ages dated to ~10,000 years old in the upper part of the core are from reworked land deposits as suggested, although if possible some additional dating controls would be beneficial to confirm the core top age. However, I don't know what relevance the following explanation about sea level has to this interpretation. Furthermore, the paragraph in one part says that the sea was close to the level of the lake at ~8 ka BP (line 172) altering the sedimentation rate in the lake, but later (at line 180-182) says the evidence supports that the lake was isolated from ~10 ka BP and at lines 218-222 that there were no storm surges after 9.5 ka BP. If there was a marine influence on the lake up until 8 ka BP this would have implications for the lower section of the record and should be discussed.

We thank reviewer 1 for this opportunity to clarify. Based on this and other reviewer comments, we sent additional dateable material to help improve our chronology. Alas, as presented in our updated age model (Fig. 2), we see a similar result: samples that are too small to be dated or/date to ca. 10,500 cal. yrs B.P. (outliers). Regarding the age of the core top, based on our experience in the field (in 2021), we are confident that we preserved the sediment-water interface (i.e. no over-coring occurred), allowing us to at least assign the year of coring (2021 AD) to the core top, despite the absence of additional dates for the past ca. 3,700 years.

Regarding any marine influence, we sought to express that the sea-level (the shoreline) was closer – rather than close – before ca. 8,000 cal. yrs B.P., as manifested by the higher observed sedimentation rates and as supported by other local sea-level studies. However, greater proximity to the shoreline does not imply that the lake had a marine influence, including storm surge imprints. Indeed, we rely on the detailed CT stratigraphy to demonstrate that the lake sedimentology bears no evidence of the latter (also see Fig. 2). We hope this explanation helps assuage the concerns of reviewer 1 regarding any marine influence.

- Table 1 – the LOI correlates most strongly with the inc/coh, bromine and Ti results, supporting my above point that organic content is an influence on the itrax data, even when normalised by the inc/coh data. The caption might be wrong: the italics in the table show insignificant results, not those greater than 0.05 probability. I think the ‘greater than’ sign is the wrong way around.

We apologize if we formatted the table incorrectly. However, we would like to emphasize that the LOI exhibits a strong negative correlation with both Br and Ti, with correlation coefficients of $\rho = -0.45$ and -0.66 , respectively. Regarding our notation of probability (p) values, p -values of 0.05 or greater are typically considered insignificant, suggesting that the calculated correlations may not be significant despite the apparent strength indicated by the p values. We hope this clarifies the reviewer's concerns about the p -value notation in the table. The italicized values, which represent weak (anti)correlations, should be interpreted with caution. We would also like to stress that as Br did not meet the applied SNR cut-offs (see the stratigraphy paragraph in our methods section) and was used as a supporting argument to our PC 1 signal, we have excluded it from the revised manuscript.

- Lines 278-280 – The point here that the strong correlation between PC1 scores and bromine supports that PC1 and bromine are both storm proxies may also be incorrect. As PC1 and the ITRAX results are potentially reflecting organic changes along the core, the bromine ITRAX results are also influenced by this issue. Furthermore bromine can be bound to organic material, therefore an increase in organic material would lead to more bromine in the sediment. This correlation with Br is also then used to suggest the results reflect the summer season when the adjacent sea is ice-free (line 281), which may not be the case if the results are mis-interpreted.

We apologize in advance for repeating ourselves here (compared to previous answers regarding the processing of our XRF data) but would like to reiterate that Br counts were normalized against inc./coh. scattering ratios, which here capture variations in organic content (reflected by the high correlation between LOI and inc./coh. in Table 1). Because of normalization, we highlight variability in Br that is unrelated to changes in organic content. We would also like to re-emphasize that all minerogenic elements in our PCA have been scatter-normalized (expressed as TSN counts), to minimize the influence of i.e. organic content after Saunders et al. 2018. However, as we merely used Br as a supporting argument, and sense that reviewer 1 is not convinced of this approach, as well as the fact that Br has not met our SNR threshold (specified in stratigraphy paragraph of methods section), we have excluded it from our revised manuscript, to help focus and clarify our approach.

- Line 288-290 – sea spray (bromine) is suggested as not being deposited when winds are too strong, as a way to explain the lack of correlation between bromine and PC2. Evidence does not support this hypothesis because sea spray deposition increases exponentially with wind strength (e.g. Franzén, 1990; Gustafsson and Franzén, 1996; Meira et al., 2008). Instead I would say the lack of correlation is because bromine and PC1 reflect organic changes, and PC2 reflects grainsize and sediment density changes.

We regret creating additional confusion and would like to re-iterate that we now chose to feature no longer prominently the (supporting) Br data in our revised manuscript but would like to stress that we do not argue that sea spray deposition decreases. We merely tried to convey that it is transported further in-land before falling out of suspension during strong winds.

- Line 305 – ‘reflect windy events or phases of stronger winds’ – not sure what the difference is? Are windy events single severe storms?

We thank reviewer 1 for seeking clarification: this is exactly the distinction we try to make, as our chronology is not precise enough to confidently distinguish between events and periods (see, for example, lines 303-305 in our original submission).

- Line 322 – the decision to choose just 2 storm records/papers is narrow, and suggests these have been chosen because of their similarity with the record presented. Papers such as Goslin 2018 extend back to the early Holocene, while the Sorrell paper only goes to 6.5 ka BP, like other more recent records (e.g. Kylander et al., 2019). A more comprehensive comparison could go in the supplementary information.

While we agree that this selection is narrow, we would like to stress that we did not base this decision on the degree of similarity. Indeed, as detailed in the first paragraph of our Holocene changes in the Easterly and Westerly wind strength section, this decision was based on an attempt to reduce (additional) noise introduced by sea-level changes (a shifting baseline) and age uncertainties, following the recommendations of Kylander et al. 2023. The former led us to exclude the highly relevant Goslin record. However, we have now gladly included it in our in our Supplementary Fig. S4. Regarding the Kylander et al. 2019 record (grain size %), we would like to stress that although it was also included in the Supplementary Fig. S4, as previously noted by reviewer 1, the Late Holocene focus of this study overlaps with the part of our record that is most poorly dated.

- Line 331 – the storm magnitude index does not add much to the paper, as the clusters of dots in Figure 5 simply align with the peaks observable in the PC records of this figure.

We regret that reviewer 1 feels that our Storm Magnitude Index (SMI) is of limited added value and agree that the information it holds is complementary. However, as another reviewer singles it out as an “interesting concept,” we decided to retain it. We ask for the understanding of reviewer 1.

- Line 353-355 – the explanation for differences between PC2 and the Sorrel storm stack record doesn't make sense, because the storm stack in Sorrel is not just based on foreshore archives, but includes sand dune reconstructions and peatland storm records too.

Here, we would like to stress that the Sorrel study is based on nine “sedimentary archives collected from open-marine settings in the macrotidal MSMB and Seine estuaries.” While these are contextualized against other storminess reconstructions, including those from peat lands (in Fig. 1), these do not underpin the analysis that identifies HSPs.

- Figure 5 – there does seem to be similar timings between the PC2 record, the Jackson peaks in aeolian activity and the Sorrel stacked record, suggesting that there are widespread increases in storminess, which is an interesting finding

Reviewer 1 is right, and this is also something that we tried to emphasize in the text, by stating that Holocene extremes of both systems coincide, which was noted too by Sorrel et al. (2012).

- Paragraph from 361 – here 5 stormy intervals are defined, so perhaps it wasn't necessary to distinguish between easterly and westerly winds? Later in the paragraph the interpretation changes back to E and W storm records again. Given concerns about PC1, I would just define storm events in the record and discuss these, or use the end member results to show changes in E and W winds.

Also following on the previous comment, reviewer 1 is right that maxima in Easterly and Westerly wind strengthen broadly coincide, and that we could fall short of trying to attribute changes to specific wind systems. However, as reviewer 1 also highlights, the information provided by the End-Member (EM) analysis of our grain size data, linked with the analyzed catchment samples (and what they tell about eolian sediment availability in the catchment), as well as the association of PC 2 with coarser and denser eolian layers (facies 2), we felt compelled to try and advance our analysis. We hope that our clarification of PC 1 (notably,

that Br counts were not included, and that each element has been TSN normalized to minimize co-variance with shifts in i.e. organic content) further allays some of reviewer 1's concerns.

- Lines 396-399 (and figure S7) – the original Fe record by Darby in figure S7c is not shown (also the IRD and MSG original records), so it is difficult to determine whether there is a convincing similarity in the variability of the Arctic Oscillation reconstruction and PC1 record. The conclusion that a 500 year lag with the 1500-year cycle in the Holocene means there was a different forcing is not necessarily correct, could it perhaps be due to chronological uncertainties between different records, or feedbacks in the climate system?

Reviewer 1 is right that we should have added the original data and is also right that – given our chronology – we should at the very least consider that this 500-yr offset might be analytical rather than environmental. However, due to no reliable chronological constraints in the last ca. 3,700 years of our record (meaning that the Late Holocene part of our record remains poorly dated), along with no proper recognition of the nature of the 500-year offset, we no longer present and discuss the dominant wavelengths (despite the fact they still follow the 1,500-year cycle), and have removed this particular figure from our revised supplement.

- The results of the spectral analysis again separate easterly and westerly winds (PC1 and PC2), which as covered above may be an issue if PC1 reflects organic variability and other drivers, such as temperature perhaps.

Based on our previous responses, we hope it is now clear that the TSN transformation of PC 1 input (which did not include Br counts) effectively minimizes any co-variance between our minerogenic indicators and organic content (due to unresolved closed sum effects). Additionally, we want to emphasize that the 1,500-year cycles reported by Bond et al. (2001) are associated with regional temperature shifts. These cycles correlate with PC 2, not PC 1.

Reviewer #2 (Remarks to the Author):

This is a very interesting paper. The rationale behind the paper is well described and reasoned. In particular, the data collected, and the statistics used have been applied in a well-thought-out manner. The resulting coherence between phases of increase wind strength with existing records from the region is a remarkable and novel finding, as is the fact that stronger Easterly winds primarily occur during colder periods. The authors argue that their data provides a link between the stronger Easterly winds and the Arctic Oscillation, but also that the Westerly winds have a different casual mechanism - even though both have a 1500-year cycle. The application

of the Storm Magnitude Index (SMI) to quantify the strength and intensity of storm is an especially interesting concept.

We thank reviewer 2 for these encouraging words, appreciating our concept of the SMI, and for giving us his credit for improving our manuscript.

The methods and data analysis are generally sound. To reliably draw the conclusions the authors have in this paper, however, there needs to be more age control. The main concern is that there are no reliable dates in the top 20 cm, covering ~4000 out of ~10,000 years of this lake record. In terms of time, nearly 40% of this record is essentially undated. This matters because the last 4000 years covers nearly two out of six (or $\frac{1}{3}$) of the 1500-year cycles that could be present in this record. To make conclusions about cyclicity of this timescale and the implications of stronger winds in general, and the Easterly SMI during the late Holocene (as shown in Figure 5a) in particular, there needs to be a reliable age constraint every 500-1000 years throughout in the record. A Pb-210 dated modern section near the top of the record with wind and storminess proxies compared to observational meteorological data would provide greater confidence in the interpretations and conclusions. There are some missing words and other minor typos in the text which are highlighted in the marked-up PDF.

We thank reviewer 2 for this comment: he is right. We would also like to apologize for pushing this (the cyclicity argument) too hard – we should have known what is spelt out here: that a significant part of our record remains poorly dated at best. Alas, although we attempted to strengthen age control for this period by extracting four additional radiocarbon samples, these did not improve our model. Problematically, the largest terrestrially sourced material seems to come from the afore identified 10,500 yrs reworked deposit (see our Core chronology, and revised Fig. 2). Finer fractions just do not yield enough material for additional reliable dates (see Table 2). So, while these additional ages strengthen the evidence that – as also suggested by previous authors (i.e. Salvigsen and Elgersma, 1993) – there is a local source of wave-reworked terrestrial material, they did not constrain the upper ca. 3,700 years of our record. As a result, and considering reviewer 1's comments about acknowledging these uncertainties (and their impact on any assessment of cyclicity), we now 1) present an age model that was run in Bacon (see our revised Fig. 2) to show more generous uncertainty margins after Trachsel and Telford (2016, in Holocene), 2) restrict CWT to the 3,700-9,700 cal. yrs B.P. part of the record constrained by radiocarbon dates, and 3) refrain from plotting the dominant wavelength(s), although they still follow ca. 1,500-year cycle, and discussing (500-year) offsets that indeed fall within the error margin of our model. We hope that these changes allay reviewer

2's concerns regarding (over)interpretation. Finally, as also visualized in Fig. S2, the upper 7 cm of this record appeared homogenized, hindering lead dating.

Comments and question from the PDF are reproduced below:

Line 167: Why did you not also use Pb-210 and/or Cs-137 dating to establish that the last 100 years had been captured at the top of the core? There is currently no age constraint for the last 4000 years in this record so all age data in this part of the record can only be considered, at best, estimates.

As mentioned above, homogenization of the upper 7 cm of our core precluded lead dating, regrettably (see Fig. S2). Regarding the (lack of) age constraints for the past ca. 3,700 years, we agree with reviewer 2. To reflect this, we have 1) re-run the age model in Bacon, which provides greater uncertainty margins (see revised Fig. 2), 2) submitted four additional radiocarbon ages, alas to no avail (although these dates further confirm a ca. 10,500-year-old source of reworked terrestrial material in the area, first reported by Salvigsen and Elgersma, 1993), and 3) reduce the emphasis on CWT and the (over)interpretation of cyclic offsets that fall within the uncertainty of our model, or remain unconstrained by ages (the last ca. 3,700 years). We hope that these changes allay the concerns from reviewer 2 that we over-interpret the Late Holocene part of this record.

Line 177: There is only one example reference included here (probably for reference limits). Nevertheless, are there any other examples because you say that it is often used. Ca has many potential sources. Is the geology as well as the setting of this example similar to your study site?

We did not add merely one reference here because of limits but agree that there are (far) more relevant references that can be cited in this context, e.g., Saunders et al. 2018. We would also like to stress that as Ca did not meet the applied Signal-to-Noise ratio (SNR) cut-offs (see the stratigraphy paragraph in our methods section) and was used as a supporting argument (Ca/Ti ratio) in our chronological part of the discussion, we have excluded it from the revised manuscript.

Line 178: This phrasing is confusing as it implies a double negative - the inclusion of the brackets is unclear

We agree and thank reviewer 2 for allowing us to clarify. Indeed, we should have just described this correlation as very weak and negative. We apologize for repetition, but we would like to

assure that since Ca did not meet the set SNR threshold, we no longer use the element or any of its ratios in the revised manuscript.

Line 216: Can you say what the other sources of Br are in your record and how you assessed the potential for Br mobility? How could Br mobility affect your interpretations when used as a sea spray proxy?

While we did correlate and normalize Br to various measures of organic content to allay concerns that Br was associated with organic matter. However, we did not investigate the down-core mobility of Br. We merely applied it similarly to, e.g. Saunders et al. 2018 and were encouraged by the correlation (ρ) with PC 1. Now, also considering reviewer comments, we question whether the supporting evidence (of PC 1) provided by Br is as supportive as we had envisioned. Also, because Br is noisier than other elements admitted to our analysis, we decided to no longer draw attention to it in our revised manuscript.

Line 309: It is good to highlight this as a potential drawback, but it leaves the reader wondering why a calibration with observational data wasn't attempted at this site. Did you collect and date sediments that overlap with the modern observational era from this site?

We hope to have conveyed that this was not possible due to homogenization of the upper 7 cm of our record, as shown in Fig. S2, and discussed in our chronology section.

Line 393: This figure is important and should be included in the main text.

Considering the aforementioned chronological uncertainties, and our attempts to address and acknowledge these, we have decided to remove Fig. S7 from our analysis. We do, however, highlight the observed cyclicity in an updated version (now Fig. S5, focusing on the dated 3,700 to 9,700 cal. yrs B.P. We ask reviewer 2 for his understanding.

Figure 3: Is Ti cps or % TSN or % cps sum? The caption says counts but these would be whole numbers if that was the case. It would be useful to also show the measured Br data, in the same way that you have plotted the Ti data. For single element plots it might be worth considering plotting these as centred log ratios (clr) to account for the closed sum effect of XRF scan data - see Bertrand et al. (2024) for further reasons.

We apologize for our unclear phrasing. For single elements, the data is shown in TSN ratios (as before), while we now apply a log-ratio transformation to ratios following the recommendations of Bertrand et al. 2024. We also attempted clr-transformation. However, the

results of this transformation were unsatisfactory (noisy). Primarily, our selection criterion for elements with 1) high sensitivity on the Mo tube, and 2) SNR not lower than 2 (Saunders et al., 2018; van der Bilt et al., 2021; see also the statistics paragraph in our methods section) encompassed only 7, before dividing their intensities by the geometric mean (Bertrand et al., 2024). Upon careful verification of this approach with Bertrand et al. (2024), personal communication with the lead author, and investigation of the seminal work on log transformation by Aitchison, we concluded that this specific approach is unsuitable for our data, as the number of elements that qualify for clr is too low (resulting in noisy data).

Line 483: Why did you not also use Pb-210 and/or Cs-137 dating to establish that the last 100 years had been captured at the top of the core? There is currently no age constraint for the last 4000 years in this record so all age data in this part of the record can only be considered, at best, estimates.

We kindly refer to our answer to this exact question at line 167.

Line 507: All of these methods would OK to apply across the whole record if the top 4000 years of the record had age constraint - as it stands, it would only seem suitable to apply these methods for the 4000-10000 year BP section which has some age constraint.

We agree with reviewer 2 and thank for his valuable suggestion. After careful consideration, we applied this method exclusively to the 3,700-9,700-year interval of our PCA data (which is constrained by radiocarbon ages), and only base discussion about 1,500-year cyclicity (which is still apparent) on this analysis, rather than comparing offsets between cycles that fall within the uncertainty of our age model.

Why did you choose 150-year moving average? Does this bias the output of the wavelet analysis in any way?

By smoothing the data with a 150-year moving average, we wanted to account for the (average) age uncertainty of our record (based on our former modelling in Clam). However, upon our revision, we no longer use this approach. Instead, we propose a 30-year (15 point) moving average, consistent with the lower 0.3 cm resolution of our physical analyses (see statistics paragraph in our methods section). We would also like to stress that we have not used the smoothed signal to perform the wavelet analysis (or any other timeseries analysis).

The supplementary material is excellent and the figures in enhance the paper substantially. It would be nice to make a summary figure for inclusion along with Figure 2 in the main text

showing, for example, one of the CT scan outputs where there is room to do this. Figure S6, the wavelet diagrams, would add great value to the main paper too.

We thank reviewer 2 for his positive assessment of our supplementary material and have added the CT slices to Fig. 2 as suggested. Considering the re-assessment of our CWT results considering the flagged (and persistent) chronological limitations of our study, we would prefer to keep these in the Supplement. We ask for the understanding of reviewer 2.

References:

Bertrand, Sebastien, Rik Tjallingii, Malin E. Kylander, Bruno Wilhelm, Stephen J. Roberts, Fabien Arnaud, Erik Brown, and Richard Bindler. 'Inorganic Geochemistry of Lake Sediments: A Review of Analytical Techniques and Guidelines for Data Interpretation.' *Earth-Science Reviews* 249 (1 February 2024): 104639. <https://doi.org/10.1016/j.earscirev.2023.104639>.

Stephen Roberts

Reviewer #3 (Remarks to the Author):

Reviewers report to “Not gone with the wind: 9,500-year sediment record of Arctic storminess favors internal climate control”.

This manuscript presents an interesting data-set from a region that is currently undergoing rapid change, and where limited previous research into paleo-storminess variability have been done. The authors have applied a multiproxy approach and analysed a sediment core from a small lake located at beach in the southern-most tip of Svalbard. The sediment sequence was analysed for grain size (Malvern 3000), chemical composition (CS-XRF) and visual variability by computed tomography. Selected proxies and their relationship were assessed through principal component analysis (PCA). The authors thereafter used principal component 1, supported by a ratio between Br and Inc/Coh ($Br/(Inc/Coh)$) as a proxy for changes in easterly winds. Principal component 2 are used as a proxy for westerly winds. Most of the figures are clearly and nicely organised.

We thank reviewer 3 for this summary, and their positive assessment of our figures.

Although the study represents a highly relevant topic and interesting data-set, it needs considerable re-working prior to publication. Some of the issues that needs to be resolved include:

1) The chronological control

The temporal uncertainties are considerable in this record, which needs to be better recognised and taken into consideration. For example, 4 out of 9 ¹⁴C dated samples showed reversed ages. After 3.8 ka (above 29 cm) all ages are reversed, meaning that there is limited (no?) chronological control after this period. The age-depth model was done using Clam, which does not depict age-uncertainty in a good way. I would recommend updating the age-depth model using R.Bacon. See enclosed example of how an updated age-model for this record would look.

Reviewer 3 is right: the uncertainties are considerable, and the upper ca. 3,700 cal. yrs B.P. remain unconstrained by radiocarbon ages, although we are confident that the core top remained undisturbed and thus reflects the year of coring (2021 AD). While there remains some debate as to whether Clam indeed under-estimates uncertainties, while Bacon might over-estimate them (see i.e. Trachsel and Telford, 2016), we agree that it is wiser to be conservative, in this respect. As such, we have re-run our model in Bacon, which results in considerably greater uncertainty estimates (800 vs. 150 years). In addition, we have sought to strengthen our chronology by extracting additional radiocarbon ages. As before, the biggest pieces of terrestrial macrofossil material exclusively derive from 10,500 cal yrs B.P. old terrestrial vegetation, whereas finer fractions were simply too scarce to yield reliable dates (see updated Table 2). So, while these results strengthen the notion first put forward by Salvigsen and Elgersma (1993) that wave-reworking of a now-submerged terrestrial (peat) deposit impacts ages in our study area, we did not manage to improve age control for the upper 3,700 cal. yrs B.P. As a result, and as we seek to highlight in more detail in subsequent comments, we have played down our major part of our CWT analysis, notably the discussion of frequency (500-year) offsets.

2) Transport processes

The site is located only 5 masl, but still only aeolian transportation is assumed. The motivation for this interpretation is not clear. Larger clasts and pebbles are also found in the deposit. The authors interpret these to be drop-stones deposited by melting ice during spring melt. If this is correct, other minerogenic material may also be deposited during the spring melt, and not via wind transportation.

We thank reviewer 3 for the opportunity to clarify the presented geomorphological and sedimentological evidence of sediment availability and transport in the studied catchment.

Yes, our lake is indeed located close to modern (and past) sea-level, and reviewer 3 is right to be mindful about the marine influence. However, as also discussed in our manuscript (see “The

Holocene evolution of Steinbruvatnet”), the investigated sediments bear no evidence of storm surges (i.e. they lack diagnostic features like erosive contacts, rip-up clasts, and even marine fossils). This is also supported by geomorphological evidence from the catchment: we did not observe surface evidence of overwash (i.e. erosion of the various beach ridges on the island). We thank reviewer 3 for raising this, as re-reading made us aware that we might not have adequately presented geomorphological evidence. We have tried to amend this, notably by adding a larger panel d to Fig. 1 that lists and shows all landforms surrounding our lake. These also include the 7-12 m high rocky ridges to the West of our lake, that likely played an important role in sheltering it from the Greenland Sea throughout the Holocene.

Regarding the referenced drop stones: we argue that these were sourced from the lake shores. On Sørkappøya, ice forming on the lake acts along the shores, rubbing and scratching the rocky western shore and the base of the uplifted marine terraces on the northern shore of the lake, and material frozen in the lake ice is delivered locally to the basin. In support of this evidence, and as further discussed under point 4 raised by reviewer 3, rounded clasts like those interpreted as drop stones by us are absent in the surrounding catchment, thus suggesting they have not been transported by wind (besides their unusually large size for eolian drift).

In addition, regarding sediment mobilization by spring melt, we would like to stress the role of cryogenic factors on sediment ability in our catchment. Frozen ground conditions and seasonal freezing bind surface sediments in our study area for most of the year (October-June), limiting sediment availability for transport to dried finer beach sediments that have not had time to freeze or have been buried under snow and/or icefoot. The modern beach and uplifted beach ridge plain are a source of only fine material, blown up from the coarser clasts after thaw or from the over-drained swash zone. In addition, as hopefully better shown by our updated geomorphological panel (Fig. 1d), there is no evidence of channels depositing spring melt runoff (and sediment) in our lake.

We hope that the above explanations help allay the concerns of reviewer 3. In addition to modifying panel d of Fig. 1, we have also added additional geomorphological observations in the revised text using track changes. Finally, we would like to point out that we have analyzed additional catchment samples (CS) 3 and 4 to better constrain the relation between eolian lacustrine input and sediment availability in the catchment (see our rebuttal under point 4, below).

3) Data handling

The processing of the CS-XRF was not conducted in accordance with the latest recommendations. See eg. Bertrand et al. 2024 for a proposed, and updated, protocol.

We would like to stress that this paper came online 2 days before we submitted our manuscript. At the time, TSN transformation seemed like the state-of-the-art of the field, based on the seminal paleostorminess paper by Saunders et al. (2018, in Nature Geoscience). We have, now, also taken note of the various recommendations proposed by Bertrand et al. 2024. Notably, elemental ratios are expressed as log ratios (i.e. $\log(Zr/K)$). In addition, we also attempted \log transformation on the selection of elements that were included in our analysis based on Molybdenum (Mo) tube sensitivity and Signal-to-Noise (SNR) cut-offs (see Stratigraphy in our methods section). However, these results proved very noisy: after consulting Dr. Bertrand, we concluded that too few variables were included. Something also flagged by Bertrand et al. 2024, as well as the seminal work on log transformation by Aitchinson. Following from the above, we still rely on TSN to help minimize the imprint of e.g. organic content on individual elements.

Also, a ratio between Br/(Inc/Coh) is used as support for interpreting variability in wind easterly wind strength. However, concentration variability of Br have been shown in multiple papers to be related to organic matter and decomposition processes, so this ratio likely do not represent changes in grain size or mineralogy which needs to be revised.

See e.g. Martínez Cortizas et al 2016 (<http://dx.doi.org/10.1016/j.gca.2015.11.013>).

We regret being unclear on the matter but would like to clarify that we use Br/(inc./coh.) not as an indicator of grain size, but as a measure of sea-spray. By normalizing over inc./coh., which – in this case, reveals a highly significant correlation with LOI: a measure of organic content, we isolate Br variability that is unrelated to the absorption of organic matter.

However, since we only used Br as a supplementary argument, and this element did not meet the applied SNR cut-offs (see the stratigraphy paragraph in our methods section), and considering that reviewer 3 is not convinced of this approach, we would like to stress that we have decided to exclude it from our revised manuscript.

The Inc/coh ratio may reflect organic matter content, but also varies with other properties, such as water content which was not take into consideration (see Bertrand et al. 2024).

While this is – generally - indeed the case, we would like to stress that, as also shown in Table 1, inc./coh. is highly correlated to LOI. In addition, we would like to stress that the relation between scattering and non-minerogenic components of the sediment that impact XRF counts

underpins the opted TSN approach applied to try and reduce this “noise” (see e.g. Saunders et al., 2018). As also noted in previous and subsequent comments, we now also use log-ratios to express elemental ratios, following the recent recommendations by Bertrand et al. (2024). We would, in this respect, also like to stress that these additional analyses have had a negligible impact on our results, as most visibly demonstrated by our updated ordination diagram (Fig. 4).

4) Interpretations

The interpretation that CP1 represents changes in easterly winds while CP2 represents westerly winds are presented in the manuscript. Unfortunately, the assumptions that these inferences are made on are highly uncertain. For example, CP1 is argued to represent easterly winds as one of the variables included in the PCA (EM1) is supposedly similar to the source sample collected immediately east of the lake, while CP1 (EM3) is interpreted as similar to the source sample collected immediately west of the lake. The results reported in (S. Fig 4) do not support this. EM1 and EM3 both show considerable overlap with the eastern beach sample (CS1), and no end member matches the western beach which is much coarser (450 μm) than EM3 (~80 μm). In addition, the beach sediments are likely a major sediment source of minerogenics deposited in the small lake, but no sample from the beaches were retrieved, instead two particular sediments were sampled 1) a silty sheet deposit (immediately east of the lake) and 2) a shadow dune (immediately west of the lake, below a bedrock outcrop). This means that the grain size composition of the most common sediment type in the area remains unknown, and thus, any west or east differences in composition are also unknown. In conclusion, it is highly unlikely that these results can be used to infer changes in wind strength in the easterly and westerly wind patterns.

We thank reviewer 3 for the opportunity to clarify our arguments regarding sediment sources and their alleged relationship with the area’s dominant wind systems – the Easterlies and Westerlies. First, we want to point out that our PCA has only been based on high-resolution XRF and CT data (see “The Holocene evolution of Steinbruvatnet”) and does not include lower-resolution physical (grain size) data. Zr/K is our PCA grain size indicator, based on previous work highlighting its potential as a coarse grain size indicator (see e.g. Davies et al. 2015, but also Cuven et al. 2011, in QSR), which is substantiated for our dataset as shown in Table 1. Based on these results, we demonstrate that coarse-grained EM 3 exerts a dominant influence on the granulometry of minerogenic lacustrine input (captured by Zr/K), which is typically dominated by fine (EM 1-dominated) silt. Here, we note that if coarsening (again,

reflected by Zr/K) would indeed result from stronger winds, rather than a different source/direction, Zr/K would align with PC 1 (minerogenic input from eolian processes, as we argue). However, this is not the case as Zr/K also shows affinity with PC 2, along with an indicator that tracks changes in grain size-related density (brought about by ensuing changes in porosity, and/or sorting – see, for example, Karstens et al. 2023, in Nature Communications) – CT greyscale.

Regarding the relationship between the EMs and catchment samples: the main argument here is that if PC 2 indeed signifies a different source of sediment, which is possibly transported by a different process (including wind direction), then we should be able to identify this with the aid of catchment samples. Here, we would first like to report that upon the revision, we have re-named our catchment samples, and CS 1 and 2 now refer to the western, and the eastern sampling locations, respectively (see Fig. 1d). For sake of clarity in this part of the rebuttal, we refer to the former nomenclature in brackets. Next, we would like to stress that we agree with reviewer 3 that two catchment samples from the direct vicinity of the lake might not tell the full story. At the same time, the location (close to our deposition centre) and context (evidence of eolian transport) do make for a compelling case. Regardless, we have now added grain size information from 2 additional samples, along a transect towards the eastern beach of the island (see Fig. 1d and Suppl. Fig. 4). As described in the revised text too, these samples derive from eolian deposits found on beach ridges (during a follow-up field campaign in 2023: this is why they were not included in our original submission). Importantly, these samples show that – while maybe slightly fining along an in-land gradient as expected for eolian input – sediment sourced from the East (with the likely exception of the actual beach, where marine processes dominate), exclusively contain the silts we associate with EM 1 in our sediments. Hence, as previously asserted, we maintain that the coarser material that dominates EM 3 has a different source. In this respect, it is regretful that we could not collect additional catchment samples along a transect to the West coast of the island: regretfully, our work in the field that day was cut short by the onset of inclement weather at an instable anchorage. We apologize for this inconvenience and are (also) frustrated by this situation. We do, however, hope, that the above evidence, and the photo in Fig. S1b showing that sample CS 1 (former CS 2) is a shadow dune deposit), strengthen our case.

Regarding the differences (and overlap) in grain size distributions that reviewer 3 mentions, we would like to stress that:

1) while EM 3 is indeed significantly finer than the shadow dune deposit of CS 1 (former CS 2), significant sorting can be expected – especially considering the coarse (sandy) grain size of this material (and hence, the energy requirements to be mobilized). As detailed in (other) grain size-based reconstructions of storminess, as well as prescribed by theoretical models, the relationship between wind speed and competence is exponential. In other words, one can, in our opinion, expect a rather steep gradient between sample CS 1 (former CS 2) and the deposition centre from which our core was taken. Indeed, the distance between the former and latter is about 1/3 of the distance between our coring site and the western beach. Again, we acknowledge that this assertion – while supported by the literature – would be stronger if only we had the time to also sample a transect towards the west from CS 1 (former CS 2). In addition, CS 1 (former CS 2) might be seen as a lag deposit – the finer component of sand from the West was re-suspended (and ended up in the investigated lake). We have now added this reflection to our revised manuscript in track changes.

*2) while the distribution of CS 2 (former CS 1) indeed overlaps with those of EM 1 as well as EM 3, its mean ($\approx 30 \mu\text{m}$) is much closer to that of EM 1 ($\approx 10 \mu\text{m}$) than that of EM 3 ($\approx 80 \mu\text{m}$). In this respect, we would like to point out that – especially in settings like Steinbruvatnet where the grain size range of sediments is comparatively modest due to a combination of available sources and active processes – the tailing of different EMs and catchment samples commonly overlap. See, for example, Vandenberghe (2013) in *Earth Science Reviews*, who report this for loess mobilized by eolian processes. We now also mention this in the revised manuscript, for the sake of clarity/transparency. In addition, as mentioned before, the additional distance between CS 2 (former CS 1) and the area where our core was extracted, can likely explain (part of) the additional observed winnowing compared to EM 1.*

I would suggest re-processing the CS-XRF data, updating the age-depth model, acknowledging the age-uncertainties, and taking a closer look to the grain size results from this record. Periods of coarsening grain size distributions may be used to infer periods of strong winds and storminess, and maybe even to infer main transport processes (see suggestions in the supplementary file).

We thank reviewer 3 again for these suggestions, and would like to re-iterate that we have indeed 1) updated the age-depth model (adding new dates, and re-running in Bacon), 2) acknowledged the much greater age uncertainties yielded by this model (for example, by executing CWT in a more cautious manner: no longer visualizing and discussing wavelength offsets that fall within the uncertainty of our model), and 3) added additional detail (and

catchment sample data) to help build a stronger case for the inferred links between sources and processes.

Reviewer 3 (in-text comments, questions, and changes)

Below, we reproduced comments, questions, and changes made by reviewer 3 and shared in pdf.

Line 29: reviewer 3 suggests following ‘knowledge’ with ‘gap’.

Agreed and implemented.

Line 30: ' The way the data is discussed and interpreted this record is mainly inferred as an aeolian signal?'

That is correct – we interpret this record as of eolian origin. Regarding the general concerns from reviewer 3 on the possible direct marine impact (of waves) on our sediment record, we kindly refer to our reply to “2) Transport processes”, where we discuss the lack of geomorphological and sedimentological evidence of storm surges (i.e. no modification of the beach ridges, and their well-preserved periglacial features, found on the island – see Fig. 1d, and no erosive contacts, rip-up clasts or other diagnostics of storm surge deposits). Accordingly, these detailed observations lead us to conclusion that both our lake record, and the geomorphology of the island bear no traces of direct marine (wave) influence, likely due to the protection offered by the various surrounding rocky ridges and skerries.

Line 33: reviewer 3 suggests changing from ‘region’s’ to region.

We improved the readability of our sentence, and now propose “changes in the dominant wind systems of the region”.

Line 34: I agree that I think that this record represents a mix of wind and wave transported material, but the conclusions in the next sentence is based on that the variability can be related to wind direction and strength changes only. It is not clear how the authors distinguish aeolian transportation from wave transportation.

We thank reviewer 3 for helping us improve the readability of our manuscript. In fact, we did not intend to make this distinction, and we have now removed ‘wave’ from the sentence. We originally, tried to convey that Br sea-spray input was ultimately wave-blown. However, as also previously stated in response to queries from reviewer 3 and others, we now dropped this proxy

due to its perceived ambiguity. In other words – our revised manuscript is focused on wind-blown transport, and we have made sure the revised text reflects this.

Line 34: reviewer 3 also suggests replacing ‘blown’ with ‘transported’.

Agreed and implemented.

Line 35: ‘Unfortunately I do not think that the data presented in this manuscript can support the conclusions presented by the authors related to 1) large uncertainties in the age-depth model, which are not taken into account when interpreting the results 2) flaws in how dominant wind direction are inferred 3) data processing, choice of proxies, and how the proxies are interpreted’

As, hopefully, also reflected by our above responses and modifications, we agree with reviewer 3 on two out of the summarized three points: 1) yes, indeed, the age uncertainties are large, and were quite possibly under-estimated by our prior Clam model (now updated to Bacon), and should certainly preclude (over)interpretation of the past ca. 3,700 yrs (now taken into account by removing part of our CWT analysis, and restricting it to the radiocarbon-constrained 3,700-9,700 cal. yrs B.P. interval), and 3) yes, while the Bertrand et al. 2024 manuscript came out 2 days prior to first submission, so that we could not take note of it in time for our first version, the information in this paper regarding log transformation, has resulted in updates to our stratigraphy figure, re-analysis of our PCA, and exclusion of the supporting argument offered by Br (previously interpreted as sea spray). Regarding point 2), we hope that the added nuance, reflection, as well as the analysis of additional catchment samples, and inclusion of geomorphological context, has allayed reviewer 3’s concerns.

Line 36: ‘I don’t agree that this is the prevalent view. My understanding is that contrasting results have been presented. See for example:

•Yin, J.H., 2005, A consistent poleward shift of the storm tracks in simulations of 21st century climate: Poleward shift of the stormtrack: Geophysical Research Letters, v. 32, p. n/a-n/a, doi:10.1029/2005GL023684

•Wang, J., Kim, H.-M., and Chang, E.K.M., 2017, Changes in Northern Hemisphere Winter Storm Tracks under the Background of Arctic Amplification: Journal of Climate, v. 30, p. 3705–3724,doi:10.1175/JCLI-D-16-0650.1.

Chang, E.K.M., Guo, Y., and Xia, X., 2012, CMIP5 multimodel ensemble projection of storm track change under global warming: CMIP5 MODEL-PROJECTED STORM TRACK

CHANGE: Journal of Geophysical Research: Atmospheres, v. 117, p. n/a-n/a, doi:10.1029/2012JD018578.'

We thank reviewer 3 for this opportunity to clarify and want to express our apology for creating the confusion. Instead, for better precision, we should have written that the view of a stormier, warmer future Arctic is “emerging” (but at the same time is usually based on very uncertain data). Upon agreeing with reviewer 3’s suggestion on the contrasting nature of presented data, we have mentioned it in the introduction and sourced from proposed papers, as well as rephrased the “prevalent” to “emerging” (using track changes).

Line 47: ‘Unfortunately, I struggled to read and understand the introduction chapter. The chapter likely needs to be revised for structure and consistency.’

We regret that the introduction was not clear and thank reviewer 3 for offering us the opportunity to improve its readability. The introduction has now been modified to emphasize the processes and factors discussed later in the article, particularly the lack of storm data in the Arctic and the role of (storm) paleodata in recognizing coastal hazards. We hope that the revised text is clearer and better aligned with the rest of the article.

Line 50: reviewer 3 suggests replacing ‘amplified warming’ with ‘Arctic warming’.

We thank reviewer 3 for improving readability of our manuscript. As the word “Arctic” is also used in the former sentence, to avoid repetition, we propose “regional warming.”

Line 57: ‘Observation of what?’

We apologize for our sloppy phrasing. We meant “observations of Arctic wave climate” and have updated the sentence.

Lines 58-59: reviewer 3 suggest replacing ‘heightened by’ with ‘in combination with’.

We thank reviewer 3 for this suggestion but propose “amplified by”.

Line 60: Reviewer 6 suggests clarifying ‘retreat rates’ with ‘coastal’.

Agreed and implemented.

Line 66: ‘So far, it sounds like this paper will be about coastal erosion, but the results and discussion are not about coastal erosion but about changes in wind direction and wind strength?’

We regret that reviewer 3 finds that the introduction is focused too much on coastal erosion. As outlined in our response under Line 47, we have taken reviewer 3’s suggestions to sharpen

its focus and consistency. Accordingly, we have modified the paragraph and reduced the focus on coastal erosion without retreating from the important link between storminess and coastal change.

Lines 78-79: ‘This sentence is difficult to read and understand and needs to be revised for clarity’.

We thank reviewer 3 for bringing attention to the poor clarity of the sentence and have changed it to: “As climate models are calibrated using observational data, their projections generally become less reliable when the variability exceeds the range of the short instrumental record”.

Line 85: ‘Lakes also receive input via fluvial input, and if located at the coast, storm and wave surges, which ref 26 used in this manuscript, also reports on. However, in this manuscript, all the particles are assumed to be aeolian transported and the motivation for this is unclear. I fail to understand how the authors came to the conclusion that aeolian transportation dominated throughout the sequence, as lakes are well known to receive input from multiple transport pathways. See, eg:

•Bertrand, S., Tjallingii, R., Kylander, M.E., Wilhelm, B., Roberts, S.J., Arnaud, F., Brown, E., Bindler, R., 2024. Inorganic geochemistry of lake sediments: A review of analytical techniques and guidelines for data interpretation. *Earth-Sci. Rev.* 249, 104639. <https://doi.org/10.1016/j.earscirev.2023.104639>

* Schillereff, D.N., Chiverrell, R.C., Macdonald, N., and Hooke, J.M., 2014, Flood stratigraphies in lake sediments: A review: *Earth-Science Reviews*, v. 135, p. 17–37, doi:10.1016/j.earscirev.2014.03.011.

* Albani, S. et al., 2015, Twelve thousand years of dust: the Holocene global dust cycle constrained by natural archives: *Climate of the Past*, v. 11, p. 869–903, doi:10.5194/cp-11-869-2015.

* Kylander, M.E., Ampel, L., Wohlfarth, B., and Veres, D., 2011, High-resolution X-ray fluorescence core scanning analysis of Les Echets (France) sedimentary sequence: new insights from chemical proxies: *Journal of Quaternary Science*, v. 26, p. 109–117, doi:10.1002/jqs.1438.*’

We hope that reviewer 3 is less sceptical about our assertion that sediment input in Steinbruvatnet is dominated by eolian processes, considering the above clarification, and data (re)analysis. In summary, we support this assertion with 1) eolian catchment samples that

display similar grain size distributions as the clastic material found in Lake Steinbruvatnet, 2) the setting of our site – proximal to the ocean, and surrounded by a small catchment without in- and outlets, or any evidence of surface run-off for that matter, and 3) geomorphological (a preserved Holocene sequence of beach ridges with mature periglacial features) and sedimentological (no erosive contacts, rip-up clasts, or other diagnostic features for storm surges) evidence that suggests no direct influence/disturbance from waves, despite the site's proximity to the ocean (and likely because it is sheltered by rocky features).

Lines 122-123: 'Please provide coordinates'

Agreed and implemented in coring paragraph in methods section.

Line 129: 'From Fig 1 d it looks it is the opposite? Ie ridges with coarser material to the east of the lake, and flat and more fine-grained material to the west and southwest?'

This is not the case and would kindly like to refer to the imagery provided in Figs. S1 and the updated panel d of Fig. 1, which provides additional geomorphological background and context.

Lines 132-133: 'How high can storm induced waves get in this area?'

We thank reviewer 3 for this interesting question. Unfortunately, we cannot give a direct answer, due to the lack of instrumental observations (although available sediment image analysis bear no evidence of waves flooding the current coastline, but (and apologizing in advance for repetitions)we would like to stress that our field observations, as well as the sedimentary lake record, enable us to postulate that these waves are (and were) effectively blocked from penetrating inland by the skerries and rocky ridges surrounding the island (Fig. 1d), and are (and were) not high enough to penetrate deep into the island. We favour the hypothesis that the rocky barrier served as sufficient shelter and wave breaker. These estimations derive primarily from the absence of overwash features or storm surge deposits in the modern coastal environment or marine fossils in our lake record. Another argument that we believe strengthens our case for no storm wave influx into the lake or significant modification of the surrounding shores is the good preservation of periglacial forms on the surface of the uplifted beach ridge plain and the development of a network of ice-wedge polygons on the marine terraces (north shore of the lake), indicating the presence of permafrost. Storm overwash or overtopping and infiltration of saline water inland would have prevented the development of permafrost. Storm flooding and inundation of the beach ridge plain would

also prevent frost sorting of pebbles on the surface of the beach ridges and the formation of sorted circles and patterned ground.

Line 135: 'From Fig 1d it looks like there is contact with another water body to the north-west?'

We thank reviewers 3 for this observation. Based on our field observations and inspection of remote sensing data, we can ascertain that there is no connection with (western) Vestre Steinbruvatnet. Two water bodies are divided by a resistant rocky ridge, which emerges from the sea and cuts across the entire southern part of the island and forms the rugged rocky lake shores not only of our studied lake but also of Sørkappvatnet, located to the south. Finally, it is worth pointing out that the other water body is ephemeral: it was present during fieldwork in 2023, but not in 2021. This is also supported by satellite imagery.

Line 160 (Figure 1c): 'Please indicate clearly the north in this figure'.

Agreed and implemented.

Line 170: 'Clam will not indicate the uncertainties of in-between dated depths and is an unsuitable age-depth modelling approach for this record. By re-doing the age-depth model with R.Bacon better appreciation of the uncertainties of the chronology of this record can be achieved. See enclosed example of updated age-model done in R.Bacon on this data.

Further reading: Blaauw, M., and Christen, J.A., 2011, Flexible paleoclimate age-depth models using an autoregressive gamma process: Bayesian Analysis, v. 6, p. 457–474, doi:10.1214/11-BA618'.

While we do not share the same level of scepticism regarding Clam, it is known that Bacon does indeed provide wider uncertainty margins. While that is not necessarily warranted (see e.g. Trachsel and Telford, 2016), we have updated our chronology using Bacon.

Lines 174-176: Since all ages above 29 cm represent reversals, and are older than depths dated between 29-79 cm, this could mean that all material in this part of the sequence could represent re-worked material. Ie, the chronological control after ~3.8 ka is extremely poor.

We agree with reviewer 3 that the lack of robust chronological control and age constraints for the last ca. 3,700 years casts valid concerns about the potential biases in our former interpretations and discussion. Thus, we express the hope that our previous answers regarding the chronological flaws and the in-text adjustments applied have provided satisfactory explanations.

We also apologize for repetition but would like to stress that while we identified and confirm occasional eolian input of the Early Holocene terrestrial plant macrofossils (ca. 10,500 cal. yrs B.P.), earlier reported by Salvigsen and Elgersma, 1993 (ca. 10,000 cal. yrs B.P.), we have not found any traces of sediment re-working, except for the excluded top 7 cm (Fig. S2). According to the observations described in “The Holocene evolution of Steinbruvatnet,” we recognize this lake record as continuous and undisturbed. In addition to the described lack of indications of storm surges after Goslin and Clemmensen, 2017 (e.g., rip-up clasts or marine macrofossils), we have also not traced any other signs of sediment reworking or different energy conditions, e.g., abrupt changes in grain size, disturbed bedding, cross-bedding or ripple marks, bio- or cryoturbations, presence of heavy minerals in secluded layers, abrupt or isolated shifts in elemental data (TSN values), irregular distribution of our physical and XRF geochemical data, colour changes (except for those that characterize facies 1 and 2). On that note, we also want to stress the lack of any signs of compaction or diagenetic processes.

Line 184: ‘The Ca/Ti ratio doesn’t seem to follow the SAR decline’.

We thank reviewer 3 for offering us an opportunity to clarify. Upon revising this information, we agree that the earlier suggested link between SAR and Ca/Ti does not look convincing. Therefore, we no more make this connection.

Lines 185-186: ‘This sentence is very difficult to understand. I guess that the authors mean that since the Ca/Ti ratio doesn’t co-vary with the minerogenic proxies, it must be derived something else? In the sentence before the Ca/Ti ratio is used a signal of marine input, but here it is stated that variability in the Ca/Ti ratio is related to the underlying bedrock? Please specify which option you will chose here, since it can not be both at the same time.’

We agree that this set of sentences is hard to follow. We do, indeed, mean to imply that – due to a lack of correspondence between Ca/Ti (now expressed as $\log(Ca/Ti)$) and minerogenic indicators, a marine origin seems plausible. However, considering the previous comment from reviewer 3, we no longer draw attention to this “argument” in our revised manuscript.

Line 189: How can the outliers and the basal depth support this statement? A few sentences above it is mentioned that the lake became isolated from the sea after 7.9 ka?

We understand the confusion, which is likely linked to the use of “emergence,” but would like to stress that we do not claim (or state, for that matter), that the lake became isolated from the

sea around 7,900 cal. yrs BP. We argue that the distance between sea and lake (and therefore eolian sedimentation rates) started to increase around this time, following the culmination of a transgression in the area that has also been reported by other authors. Hence, the presence of terrestrial radiocarbon-dated material with ages as old as ca. 10,500 cal. yrs B.P. suggests that the surrounding area was isolated much earlier than previously thought (see i.e. Forman et al. 2004). We have now modified the revised text to better reflect this line of argumentation (see track changes), and phrase it more clearly.

Lines 215-216: ‘This does not represent the most updated protocol to process CS-XRF data. Please see and follow Bertrand et al 2024 for an updated protocol to process CSXRF data’.

We kindly refer to our response to reviewer 3’s major comment 3 “Data handling”. We have taken note of – and now follow – the recommendations provided by Bertrand et al. 2024, which appeared online 2 days prior to first submission of our manuscript.

Line 223: ‘Br is associated with organic matter content, and also decomposition processes, and may thus not be a good sea spray proxy. See eg: Martínez Cortizas et al 2016 (<http://dx.doi.org/10.1016/j.gca.2015.11.013>)’.

We hope that by addressing this topic under "Data handling", we have convinced reviewer 3. In short summary: although we did normalize Br over inc./coh. ratios – in this study, based on a strong correlation with LOI (see table 1), a proxy for organic content, we did drop it from our story, as it was merely used in further support of PC 1, and raised more questions than answers by reviewers (also other than reviewer 3).

Line 228: ‘There are big clasts in the sediments which may point towards other transport processes. Was the sediment analysed for marine fossils such as shells and/or diatoms? If so how? (not mentioned in the manuscript).’

For an elaborate answer, we kindly refer to our response to major comment 2) by reviewer 3. In summary, we argue that the absence of storm surge deposits (no erosive contacts, or other unconformities visible on the CT imagery, and preservation of an 8,000 yr old sequence of beach ridges as well as mature permafrost features preclude any direct marine (wave) influence. In addition, cryogenic processes, and the lack of any channel-like features disfavour significant surface run-off (due to spring melt) in our small catchment. Finally, clasts of this size and (rounded) shape are found on the western beach of investigated Lake Steinbruvatnet.

Hence, as is common in Arctic lakes, we argue these large clasts are drop stones deposited from lake ice. As mentioned at line 221 of our original submission, we found no marine fossils in our material. This conclusion is based on 1) light microscopy, to identify and isolate microfossils for radiocarbon dating revealed no foraminifera, and 2) CT visualization, which revealed no (broken) shell remains.

Line 240: ‘Couldn’t this indicate that the lake receives melt water inputs during spring? And that this is the main signal? Please also consider if the lake area can vary by season.’

We kindly refer to our response to one of reviewer 3’s questions under major comment 2) “Transport processes”, where we also underline no evidence of channels depositing spring melt run-off (and sediment) in Lake Steinbruvatnet. Moreover, we would like to point out that spring snow and/or ice melt does not preclude eolian origin of these sediments. Finally, we observed no evidence of past changes in lake water level (i.e. raised lake beaches or wave-cut notches).

Line 243: ‘See previous comment and Reviewer report regarding this assumption’.

We hope that that discussing this matter in more detail under the “Interpretations” major comments, along with the extra grain size analyses carried out, has presented reviewer 3 with more compelling arguments.

Line 244: ‘All grain sizes influence the grain mean size’.

We thank reviewer 3 for the opportunity to clarify. We agree with the comment and have changed this part of the sentence to: “In contrast, sand-dominated End Member (EM) 3, which exerts a strong influence on grain size and also co-varies with coarse grain size indicator Zr/K [...]”.

Lines 245-246: ‘The grain size distribution of EM3 and CS2 are not nearly identical. The mode of EM3 seems to be around ~80µm, while CS2 (“western beach”) has a mode around ~450µm’.

Reviewer 3 is right, and we apologize for phrasing this so brazenly. In our revised manuscript, we now tone this down, to better reflect the contents of our reply under the major comment on “Interpretations” by reviewer 3. In short summary, while we acknowledge that the grain size mean of EM 3 is finer than that of CS 2, we also stress that eolian sediment of this grain size (sand-size) is exclusively found towards the west of our lake basin. In this sense, we would like

to draw attention once more to the added catchment samples (also see Fig. 1d). Finally, we would like to emphasize the well-documented reduction of grain sizes with distance (as our coring location lies much further down wind than CS 2) and raise the possibility that CS 2 is a lag deposit (finer material like that comprising EM 3 has been re-mobilized by the wind). We hope that these arguments allay the concerns from reviewer 3.

Line 252: ‘If the pebbles are deposited during spring melt as drop stones, maybe the other material can be too?’.

We kindly refer to our response to one of reviewer 3’s questions under “Transport processes”, as well as to our response to reviewer 3’s question under Line 240.

Line 277: ‘This ratio varies with organic matter content, but is also affected other factors such as water content (See e.g. Bertrand et al 2024). The LOI and Inc/coh ratio r value is around 0.7 in this data, meaning that c 50% of the variability is shared’.

Indeed, about 50% of variance is shared between both variables, which is considered a strong correlation by most (geo)statistical resources. Especially when keeping in mind that the much higher resolution inc./coh. had to be resampled to allow direction comparison (see statistics paragraph in our methods section). That said, it is possible that other factors like water content also influence inc./coh. ratios. For a more elaborate reflection on this matter, and other issues related to XRF data processing, we kindly refer to our response to the major comment on “Data handling” by reviewer 3.

Line 278: ‘The correlation between MSG and the Zr/K ratio is quite weak (0.4), please revise’.

We agree with reviewer 3 that a correlation coefficient of 0.4 is not strong but disagree that it is quite weak. In line with most geostatistical reference works, we now refer to this correlation as moderate. We ask for reviewer 3’s understanding.

Lines 279-280: ‘This sentence is difficult to understand, please revise’.

We thank reviewer 3 for offering us an opportunity to clarify. We have now changed the sentence to “We selected XRF elements with 1) high sensitivity to the fitted Mo tube, and 2) a Signal-to-Noise ratio (SNR; μ/σ) higher than 2 after Saunders et al. (2018), van der Bilt et al. (2021)”. On that note, and as written in our former responses to reviewer 3, we have not included Br and Ca in the revised version of the manuscript.

Line 282: ‘This association is likely related to that EM1 represents silt sized grains, which are the dominant grain size. Meaning that when there is higher minerogenic content the color will change, and also the density.’

We are unsure about the context regarding colour of this comment but would like to stress that – generally speaking – minerogenic material is denser thus lighter on CT scans (higher CT greyscale values). While there is certainly more to it (i.e. related to sorting and grain size), this is one of the reasons for using CT scans in studies like these. We hope that this response adds information to reviewer 3’s statement.

Line 283: ‘See previous comment about the limited source samples, and also interpretation of grain size data. Both EM1 and EM3 overlap with the sample immediately east of the lake. Thus, the interpretations that follow here (that EM1 represents the polar easterlies) is not supported by the results presented here’.

We hope that by covering this issue extensively in our former replies to reviewer 3’s questions, particularly under “Interpretations”, we have offered a more compelling argument. In short summary, whilst acknowledging that our wording regarding the agreement between EMs and catchment samples was too strong and unnuanced, we also stress that it is typical that the grain size distributions of distinct EMs overlap. Differences in means/modes is what matters most here. And, in that respect, EM 3 is much more similar to the sands of CS 1 (west) compared to the silts of CS 2 (east). In addition, we also make the case that the offset between CS 1 and EM 3 might be explained by the transportation distance from the source (western beach) as well as progressive winnowing of CS 1 (leaving the coarsest fraction that cannot be mobilized by wind). Finally, we add additional eolian catchment samples along a transect to the eastern beach (see Fig. 1d) that – in our opinion – convincingly demonstrate that the silts that characterize EM 1 dominate the eastern shores of Steinbruvatnet (there is nothing as coarse as EM 3 on this side of the island; see also the Supplementary Fig. S3).

Lines 293-294: ‘See previous comments about sampling, processing and interpretation of the data’.

We would kindly refer to our earlier replies on that topic, particularly those provided under the major comment on “Interpretations” raised by reviewer 3.

Line 305 ('Holocene changes in Easterly and Westerly wind strength'): 'Due to the uncertainties in the age-depth modelling, processing, and interpretation of the data these inferences can not be reliable be made. Please revise.'

Referring to our responses to the (numbered) major concerns raised by reviewer 3, we 1) agree that the uncertainties of our chronology were not adequately considered in our first submission, compelling us to re-run the age model in Bacon, and reduce the scope and ambition of our CWT efforts to the radiocarbon-constrained 3,700-9,700 cal. yrs B.P. interval, 2) agree that our data processing could be improved, compelling us to do so (notably, by including additional catchment samples, but also by following best practice for XRF data processing after the study by Bertrand et al. 2024, which was published 2 days prior to submission of our original manuscript, and 3) hope that – in light of the above – the confidence of reviewer 3 in our – more conservative interpretations – has been strengthened.

Line 348: 'I do not agree that there is a striking resemblance? Please also outline the differences and similarities in study approaches of the studies that you compare with, so that the reader can get a sense of what factors that may influence the results.'

While we would like to stress that we argue that the similarity is striking in the context of notoriously noisy storminess records (as discussed in this section too), we have now removed "striking" to accommodate reviewer 3. In addition, we have added a bit more detail regarding the approaches and sites (notably setting, proxy, and sea-level history) used by the studies we rely on for comparison – see the track changes in our revised manuscript.

Line 350: 'The data from the Jackson et al (2005) paper uses the mean grain size of their data to infer changes in wind strength whereas this manuscript takes a completely different approach. Since you also have grain size data, it could be relevant to compare the mean grain size of this dataset compare to Jackson et al record'.

While our mean grain size data show a similar pattern to those presented by Jackson et al. (2005) - see below, we argue that our other eolian proxies add valuable information, and that these gradients of variability are best captured by PCA, following the seminal paleostorminess work by Saunders et al. (2018) in Nature Geoscience.

Lines 369-370: How are these periods defined? For example, from your results the 5.5 ka “event” looks quite long? The same with the 7.4 “event,” this looks like it spans 6.5–7.5 ka.

We thank reviewer 3 for this question. Frankly, our approach was rather qualitative, and based on similar previous studies, by stating the age on which any phase (we refrain from using “event” or “period”) was centred. Now, in our revised manuscript, we explicitly identify these along the same lines we build our SMI, and state clearly that the numbers listed reflect the mean age on which any phase – when storminess (PC 1 or 2) exceeds the mean + standard deviation cut-off – is centred. We hope this allays the concerns of reviewer 3.

Line 370: ‘These two studies identified cool periods based on distinct papers and study approaches. From what I can see, the cold periods of these refs do also not agree? I would advice doing this comparison in a more careful manner. For example Wanner et al. (2011, <http://dx.doi.org/10.1016/j.quascirev.2011.07.010>) defines cool and dry periods. How would your results compare with these cold periods?.’

We thank reviewer 3 for suggesting the Wanner compilation – which is based on a great number of regional record and therefore, arguably, more representative. At the same time, we would like to stress that both Jackson et al. (2005) and Sorrel et al. (2012) studies primarily rely on the work by Bond et al. (2001) to define their cold periods. Regardless, we once again thank

reviewer 3 for suggesting comparison against the Wanner compilation and have included it in our revised Fig. 5.

Line 386 ('Wind extremes track phase-lagged 1,500-year climate cycle'): 'See previous comments on issues with chronological control, data processing and interpretation'.

For a more elaborate response, we kindly reference our response to the major comments raised by reviewer 3. In short summary, we agree, and have significantly toned down any remaining CWT work. We fully agree that – given the average 800-year uncertainty of our updated Bacon age model – we cannot confidently discuss lags between 1500-year cycles.

Suppl. S4: 'I would recommend presenting the grain size results in more detail, such as 1) reporting which grain size that is most commonly occurring, reporting, and plotting sorting and skewness and the different grain size distributions (clay, silt, sand etc) throughout the record. Followed by a presentation of the end member results and what the different end members represents etc. Maybe the different end-members presented in Fig. S3 represents different transport processes or re-working (see e.g. Vanderberge et al 2013), such as fluvial input during the spring (EM1?) and aeolian transport and deposition (EM2 and 3?). Please also consider how many years each analysed sample represents. Given the coarse resolution of this record, it is likely that each sample represent multiple years, transport processes and seasons. From the data I can see that the silt is the dominant grain size here, which is in the end member modelling represented by EM1, while sand is a minor component (EM3) which might be important for the interpretation.'

Regarding the discussion on what our EMs might reflect, we kindly refer to previous queries, notably our answers on reviewer 3`s major comments about 4) Interpretations. Regarding the time covered by each sample, we`d like to stress that – as also articulated from line 302 and onwards - that the uncertainty of our chronology hampers such an effort: we cannot be certain whether the detected storm facies reflect individual events, or phases. We ask for the understanding of reviewer 3. We thank reviewer 3 for the suggestion to add additional grain size information and have complemented our supplementary data file with these metrics.

We would also like to mention that, upon the revision, we now present the EM 1-3 and CS 1-4 data in Supplementary Fig. S3 (former S4).

We thank you and the reviewers for this opportunity to help further improve our manuscript. In the text below, we address each raised point (in italics). Summarizing what is stated there, in relation to the two major points singled out (PCA and regional representativeness), we:

1) Show that the correlation between PC 1-2 that reviewer #4 reports appears to stem from a spreadsheet calculation error: when including the first 520 data values up until the first data gap in the PCA tab of our data file, one indeed derives a Pearson correlation coefficient (r) of 0.55. However, when including all ($n=4729$) PC 1-2 combinations, we derive an r value of 0.000. Hence, we re-affirm that our PCs are completely independent from each other. Regarding the other points raised by reviewer #4 about our PCA analysis, we clarify that clustering of samples around the centre of our ordination diagram merely reflects a short length of the environmental gradient(s) captured, rather than a concern about the “trustworthiness” of the analysis. Moreover, we emphasize the difference between variance and significance in relation to our PC 2. This PC indeed captures a modest percentage of the total variance of our dataset, as just 2 out of 6 variables cluster along this axis, but that does not make this signal insignificant. Indeed, PCA is often used with the explicit purpose of teasing out such “hidden” signals using secondary PCs (we provide an example below).

2) Further acknowledge that our findings are indeed site-specific. While characterized by a unique set of boundary conditions (stable sea-level, facing both of the region’s dominant wind system, and located around the polar front zone where these meet) that make our study site particularly suitable for a paleo-storminess reconstruction (see our Setting paragraph), and recording a signal that is broadly consistent with other reconstructions of Westerly and Easterly wind strength from the North Atlantic region, our dataset is the first of its kind from the Arctic. We therefore agree with reviewer #4 that additional studies are needed to confirm the representativeness of our dataset. We apologize for not clarifying this (sufficiently) in earlier versions of this manuscript, and have addressed this by repeatedly making this point at visible locations in our latest revision. See for example, lines 92 (the final sentence of the introduction), and 428-429 (the final sentence of the discussion) in the version with track changes. In addition, we have also modified the title slightly to better reflect this notion (and the summer seasonality of our proxy evidence).

REVIEWER COMMENTS

Reviewer #2 (Remarks to the Author):

I have read the revised article and their response to reviewers report, which is very thorough and comprehensive. The authors have undertaken a substantial revision to the previous submission and have addressed all previous concerns in a sensible and realistic manner.

We thank reviewer #2 for this positive assessment.

Overall, their more cautious interpretation of the results in the paper is admirable, and, in my opinion, does not detract in any way from either the significance or novelty of their findings. In particular, I am pleased to see that they now address the shortcomings in the record's chronology and the impact that this has on establishing cyclicity in parts of their record that are not well-dated, i.e., the last 3,000 years. Their approach to dealing with Br also makes sense, as this element is notoriously difficult to interpret in terms sea spray from ITRAX data alone, especially where large shifts in organic content exist downcore.

We thank reviewer #2 for these kind words, and hope to find new ways to better incorporate halide ion data in more meaningful ways in future work.

One additional (minor) comment on the revised submission, which **they may wish to consider**, relates to the use of %TSN for Ti and how the authors dealt with noisy clr transformation. Although some previous papers use the %TSN method (and the guidance from the Cox analytical is it remains a valid normalisation technique for taking variations water and organic content in scan data into account), it is probably worth noting that %TSN produces the same result as calculating the % of cps sum. The latter is mathematically more straightforward and understandable, mainly because it is analogous to transformation of species count data to percentage datasets. The %cps sum is also described as a possible data transformation method in our recent Bertrand et al. (2024) paper. My personal preference these days, following final recommendations in the Bertrand et al. (2024), would be use clr transformed datasets when dealing with individual element profiles. As the authors discuss in their response document, there are some drawbacks with the clr method, mainly that it produces very noisy element profiles when some elements contain many zero cps datapoints. Clr and log ratios both involve the use of log transformation of elements with zero values, which is especially problematic for clr as multiple elements are incorporated into the calculation and because the whole row is then lost if any of the elements contain zeroes. However, it is possible, and statistically valid, to replace zero values with either half each elements' minimum value or by a value that is a randomly generated fraction of the detection limit for each element prior to clr (or log ratio) transformation. While zero replacement should be considered carefully for each individual

dataset, zero replacement will enable the retention as much non-noisy element data as possible (i.e., elements without many zeros). The clr process and zero replacement can be undertaken rapidly within R and/or using the 'zeroreplace' function in the 'Compositions' R package - it would be a good idea to read the caveats in the package notes before proceeding with this. Bertrand et al (2024) also recommend using incoherence normalised log ratios when investigating individual elements, in this case $\ln(\text{Ti}/\text{inc})$, for reasons outlined in the paper - this approach would be more consistent with the reasoning the authors use elsewhere in the revised submission wrt Zr.

We would like to clarify that our clr results were not noisy due to zero values, but because our selection of (n=6) elements (following the application of tube sensitivity and signal-to-noise criteria as detailed in the methods section) was close to the minimum number (n=5) of elements required for this transformation – as stated by Bertrand et al. 2024 (and based on the pioneering work by John Aitchison). After consulting the lead author on this matter, with the intent to normalize our data and also reduce the imprint of matrix effects, we settled on TSN transformation for Titanium (Ti) after Saunders et al. 2018. We had not considered normalizing by incoherent scatter as reviewer #2 suggests, because the afore-mentioned Bertrand study describes it as “inappropriate” after Croudace and Gilligan (1990). We have, however, followed the recommendation of reviewer #2 to express Ti as $\ln(\text{Ti}/\text{inc})$ and find that this result is near-identical ($r=0.96$) to our Ti TSN curve (data available on request). Based on this striking similarity, we decided to retain the TSN results (input to all subsequent analyses). We ask for the understanding of reviewer #2.

Reviewer #4 (Remarks to the Author):

The authors present lake core data from the southern tip of Svalbard, Sørkappøya island, from which they conclude that “a colder Arctic is stormier.” The island has a unique location in the sense that it appears to preserve an interesting record of aeolian deposition during summer months. The main conclusion of the paper is that “we reveal quasi-cyclic wind maxima during regional cold periods “...(which) challenge the emerging view that a warmer and less icy future Arctic will be stormier.” Unfortunately, the main conclusions are not supported by the data, due to the following three points.

1) As mentioned by the authors, the lake is covered by ice during 9 months, and is therefore only receptive to direct aeolian deposition during a few summer months. However, storms are most intense and frequent during winter. Therefore, the data can not be used to assess general

“storminess”, only storminess during summer, and the main conclusion is thus not supported. In the discussion, this seems to be overlooked and the results are compared to annual storminess from other records (see the Iceland and Outer Hebrides studies).

We agree with reviewer #4 and would like to stress that this point is made: see lines 340-341, and also line 483 in our first revision (with track changes). To accommodate reviewer #4, and further clarify that we by no means contend that our record captures general storminess, we now elaborate more on the seasonality of our reconstruction in the discussion (in track changes). Regarding the Iceland (Jackson et al. 2005) and the Hebrides (Kylander et al. 2020) studies: the former is skewed to the summer for the same reason as our reconstruction – sub-Arctic Iceland soils are also frozen and snow-covered outside this season, so that sediment cannot be mobilized, while the comparison against the latter (the Hebrides) was added in response to a reviewer request – the reason it is part of a supplementary figure, rather than featuring in the main text, is because it did not meet our key selection criteria (see lines 382-383 in our first revision). We hope that this clarification allays the concerns from reviewer #4.

a. Why is the relationship between summer and winter storminess not even discussed?

As outlined above, and also stated in the Holocene evolution of Steinbruvatnet section of our manuscript, we are confident that our record captures a summer signal. Without data, any discussion of the relation between summer and winter storminess in Steinbruvatnet would be hypothetical. To overcome this limitation and address the point reviewer #4 raises, we now discuss a very recent compilation of modelling simulations, which suggests that future changes in storminess are greatest in winter (lines 399-401).

b. How did lake ice coverage change throughout the Holocene, and how does this influence the storminess reconstruction?

We thank reviewer #4 for raising this important point. In the absence of site-specific information about Holocene changes in lake ice cover, we have to rely on secondary information. Specifically, paleo surface temperature data from the surrounding Barents Sea (see Fig. 5k): the main control on the duration/extent of lake ice cover. While the Early and Middle Holocene were comparatively warm, and the duration of snow- and ice-free conditions therefore likely longer, this is not reflected in an increase of wind-blown sediment. We therefore argue that changes in lake ice were either minor, and/or did not significantly impact sediment fluxes into Steinbruvatnet. Our revised manuscript now reflects this line of argumentation (see lines 391-393 of our second revision).

- c. In line 404, the authors state that “our data is biased towards summer”, but really the data only relates to summer storminess.

We agree, and have modified the text accordingly.

2) Presenting data from one site in the Arctic is not enough to make conclusions about reconstructing “quasi-cyclic wind maxima during regional cold periods” and to evaluate whether “a warmer and less icy future Arctic will be stormier”. Throughout the manuscript it is implied that the data is representative for ‘the Arctic’ (what region do the authors mean by this? The whole Arctic? Svalbard?). To support the claim, the community will really want to see how reproducible these results are, for example ‘across Svalbard, before a claim regarding the Arctic can be accepted. In NW Europe, it has proven difficult to reconstruct general patterns of overall storminess, simply because climate change can cause storm tracks to shift. Would this be different in the Arctic? Any discussion on the representativeness of the site for ‘the Arctic’ is entirely lacking, as well as a discussion on potential storm track shifts, presenting a major flaw.

Reviewer #4 is right: we should be mindful that – albeit from a strategic location – our study and its findings derive from one site. While showing broad agreement with existing storminess North Atlantic reconstructions from areas further South (see Figs. 1 and 5), whether or not our reconstructions are representative for the study area (Svalbard), or region (Arctic), will require additional data (as our record is a regional first). In our second revision, we now address this head-on by 1) clearly distinguishing between our study area (Svalbard) and the wider region (Arctic), and 2) stressing the site-specificity of our findings, and arguing that – while broadly consistent with existing/previous North Atlantic work – any statements about their representativeness for the area/region requires more work (as our study is the first storminess reconstruction from the Arctic). We also make a specific suggestion, following the recommendations by Kylander et al. (2023), by recommending that future data should be collect a transect of sites, to enable capturing latitudinal wind shifts (see lines 429-431).

Regarding the point raised about the position of storm tracks – our proxies merely pick up on changes in wind strength (transporting more or less clastic material). Alas, we cannot disentangle to what extent these are driven by a general increase in wind speed, or latitudinal shifts of storm tracks. In this respect, we would like to stress that most paleo storminess studies use both inter-changeably (see, for example, the Sorrell et al. 2012 synthesis, but also the Saunders et al. 2018 study, both in Nature Geoscience). To accommodate reviewer #4, we now

explicitly state that we cannot attribute changes in wind strength to windspeed and or storm track position (see our revised “Holocene changes in Easterly and Westerly wind strength” section -lines 319-321), which would require a transect of sites (see lines 429-431).

3) The authors have solid and interesting primary data (grain-size analysis, end member modeling, XRF data), but they heavily base their interpretation on a derivative of the primary data, relying on PCA analysis. Few arguments are presented why the PCs are interpreted the way they are, and I am concerned about the trustworthiness of the PCA analysis due to the following reasons:

a. The variability of the PC2 looks spuriously similar to PC1. This is odd because PC2 is supposed to capture the ‘leftover’ variability that is not captured by PC1. I quickly checked the data and the pearson correlation coefficient between PC1 and PC2 is 0.55 – this is way too high and likely points to artifacts created by the data analysis/preparation. I recommend the authors to thoroughly check their PCA analysis.

Indeed, individual Principal Components (PCs) should – per definition – capture different gradients of change in datasets. This is also the case with our Steinbruvatnet data. Reviewer #4 states the data was “quickly checked”, and we fear something might have gone wrong there. When thoroughly checking our PCA (PC 1-2 scores) data, we found a non-existing correlation of 0.000 when assessing all 4729 data combinations. However, when using the ctrl + shift + down arrow short cut key in Excel, only the upper 520 values are selected, as subsequent cells are empty (no data). This sub-selection of ~10% of the PCA data indeed yields an r value of 0.55. To avoid similar mistakes by other users of our data in the future, we have now removed the empty (no data) cells in our PCA data summary. We trust this allays reviewer #4’s concerns.

b. In the PCA biplot no clustering of the sample scores is visible, they all sort of plot around the origin. Therefore it is quite unconvincing that the PCA is really capturing underlying structures of the data. It is OK to try PCA on a dataset to see if patterns can be discerned, but one should critically evaluate the results. It should also be noted that PC2 only captures 11% of the variation.

We are unsure why grouping of samples around the origin of our ordination bi plot makes for an “unconvincing” PCA. This data structure merely suggests that our dataset is comparatively homogenous (which is the case – as there are no specific and distinct units, just semi-regular alternations between facies 1 and 2), and therefore encompass a short environmental gradient. See for example: https://www.davidzeleny.net/anadat-r/doku.php/en:mix:ordination_artefacts.

Even more so, and also in light of the previous concern from reviewer #4 about the independence of our PCs: in cases where PCs really are related, one would see a characteristic curvilinear sample distribution – the so-called Arch or Horseshoe effect. As can be seen in Fig. 4, our PCA results bear no evidence of this artefact. We trust that this explanation helps allay the concern of reviewer #4 about the structure of our ordination diagram. Finally, it is true that PC 2 captures “only” 11% of the variance captured by our dataset. But this does not mean that this signal is insignificant. It merely arises from the fact that two out of six of the included parameters record a signal that (also) aligns along (have loadings on) this axis. We would also like to stress that PCA is often used with the explicit purpose of teasing out such hidden/overprinted secondary signals. A notable example, in a lake-sediment-focused Arctic paleoclimate context, is the seminal work by Olsen et al. from 2012 in Nature Geoscience: here, PCA 3 (explaining ~9% of dataset variance) tracks North Atlantic Oscillation (NAO) change.

c. The most direct proxy for wind strength, i.e. actual physical grain-size measurements, are not taken into account in the PCA analysis. It would be much better if the authors make their case based on the primary data, rather than an unconvincing PCA analysis. As a result, the PCA-derived SMI index is also strongly jeopardised because of this.

While the physical grain size measurements are not part of the actual PCA, owing to their much lower (0.3 cm vs. 0.02 cm) sampling resolution, we do substitute grain size information with a derivative: as explained in our Holocene evolution of Steinbruvatnet section, the widely used (see i.e. Davies et al. 2015) particle size proxy $\log(Zr/K)$ capture 40% of mean grain size variability. As also outlined in our manuscript (the Holocene evolution of Steinbruvatnet section, and especially lines 315-320, but also Table 1 and Fig. 3), facies 2 (wind input) horizons are (far) too thin to resolve with physical samples, so we combine (and substitute) these with high-resolution scanning approaches. Regarding the PCA results: we hope that our previous explanation about the interpretation of these data, and clarification of the calculation error that might have led reviewer #4 to suggest that our PCs were not independent, has allayed concerns about the strength of this analysis.

4) I share the concern with reviewer 3 that if cobbles and pebbles can be ice-rafted, then why not some of the fine sand? The authors know the study area best, but reading the response letter it is still not clear to me whether (fine) sand is presented in the vicinity of the lake that could be rafted to the basin. Are the authors willing to state that “there is no fine sand present around the lake that could be transported by lake ice into the lake” (if it is true)?

We thank reviewer #4 for raising this, and now see that we might not have adequately addressed this point following the previous round of revision. We are indeed willing to state that (presumably as a result of wave action) there is no fine sand present around the lake shores, and now do so in the revised manuscript. Only pebbles are therefore available for ice rafting.

Other comments

-A map of modern storm tracks is really missing.

We have now added schematized arrows to indicate the dominant influence of the Westerlies and Easterlies in our study area, also highlighting the position of the polar front at their interface (after Parsons et al. 1996: see Fig. 1). Because of the significant spread/variance of modern storm tracks (see for example <https://coast.noaa.gov/hurricanes>), and concerns about their representativeness for the multi-centennial resolution paleoclimate perspective of our study, we have refrained from adding individual storm tracks recorded in our study area over the (short) instrumental period, in line with other paleo storminess studies.

-Please provide a tradition clays/silt/sand diagram in Fig. 2/3. Please provide a litholog and a plot showing mean grain size changes.

Agreed and implemented (as supplementary figure S3). We would like to stress that mean grain size changes are already plotted – see Fig. 3: MGS. In the same figure, we also show a log that marks the alternations between facies 1 and 2 after Hess et al. 2023 in JQS.

-Captions of the Supplementary figs are missing?

We regret hearing that reviewer #4 could not see the captions of the supplementary figures, and checked our submission in the journal's portal. Here, as well as in the file that we submitted, they were present and readable. Could it be that the un-captioned figure files that reviewer #4 refers to, are those appended at the base of the manuscript? If so, we would like to point out that these are main text figures, that are integrated in the manuscript (captioned).

Reviewer #5 (Remarks to the Author):

The ms, based on a lake core from a strategic location at the southern tip of Stitsbergen, gives a unique multi-millennia scale record of Holocene changing wind-systems over the Arctic and North Atlantic – a key area in the debate of Global change. As such, it will contribute to the rapidly growing literature on the importance of storminess in future climate change. Contrary

to some earlier literature, the record indicates that future warming should not increase storminess in these parts.

The c. 1 m lake core is made up of eolian silt, blown in from both east and west, and treated with an array of sophisticated instrumental and statistic methods to distinguish source eas and wind intensity. I am not an expert in these fields and will not comment on them, although they are a very important part of the work. Hopefully others will take this up. (The authors appear to me as competent and with a sound critical approach to their results). All this evidence is interpreted in terms of the local coastal morphology and sedimentation.

We are humbled by the assessment of reviewer #5 of our work.

In the discussion the results are compared to other multi-millennial scale records from coastal NW Europe, finding a broad agreement with these records with cold periods being stormy, although there is some disagreement about the wind situation during Holocene warm period (see f.i. the recent paper by Sjöström et al., QSR 2024).

We thank reviewer #5 for bringing this manuscript under our attention, we had regrettfully missed it before. This is an impressive piece of work, and indeed fits well with the narrative of a stormier Late Holocene. It deserves to be cited here, and we do so now. Regarding the Early Holocene bit of this record, it is interesting to observe that the authors record no storm periods prior to 6 ka BP (although this might be related to the concurrent sea-level low stand).

In general I find the ms well written with relevant and well argued points, but I do have a few suggestions for the authors to consider.

The role of lake ice is referred to only sporadically, but deserves a closer look. Presently the lake is ice-free for only three months of the year (which may explain why only c. 1 m sediment has accumulated over c. 10,000 years). Therefore, the record is strictly a summer record, leaving out the winter, which is usually the time for storminess. This should be considered, not least when comparing to lake records from more southerly areas where lake ice is absent. Also, the annual duration of lake ice must have varied over the 10,000 years. An increase or decrease of one month per year, which is not an unreasonable range, would prolong or diminish the window available for sedimentation with 25%, with consequence for sedimentation rates, independent of such factors as proximity to the coast.

Reviewer #5 is right: we argue that our proxy reconstructions capture a summer signal, but only do so in the Holocene evolution of Steinbruvatnet section of our manuscript. We now make

this important point more explicit, by also mentioning it up-front in both abstract and introduction, while also stating in the discussion that recent modelling work suggests that future Arctic wind changes are greatest during non-summer seasons (fall and winter, in particular). In addition, to allay concerns that changes in snow and ice cover significantly impacted sediment mobilization and deposition during warm periods, we now highlight that inferred eolian input exhibits no relationship with the temperature records shown in Fig. 5 (line 391).

I suggest that the local RSL curve, which is an important parameter and frequently referred to, is reproduced together with the age model in Fig. 2.

We would like to have done so, but have refrained from doing so, for two reasons. First and foremost, this curve is based on just two tie-points: one around 3 ka BP (0.2 m a.s.l), and the other around 7 ka BP (9.5 m a.s.l) – see Ziaja and Salvigsen (1995). Moreover, both these samples encompass marine biomass (seaweed and whale bone) that is impacted by marine reservoir effects. Subsequently, Forman et al. (2004) extrapolate sea-level rise further back in time in their compilation, but without the inclusion of additional data. Additional information is provided by Salvigsen and Elgersma (1993), who date chunks of peat that have been wave-scoured from the now-submerged shallows between Sørkappøya and mainland Spitsbergen to around 10.5 ka BP. However, while indicative of a low-stand, these shallows are of a variable 3-9 m depth, and can therefore only loosely constrain sea-level at the time to 5 m below today.

Finally, all of these constraints derive from Sørkapp Land, not the island (Sørkappøya) itself. In light of the above, we argue that the existing (rudimentary) sea-level curve only adds noise/confusion, due to the outlined combination of data paucity, spurious extrapolation, and site-specificity. We ask for the understanding of reviewer #5, and hope to return to Sørkappøya and construct a new curve, based on paired wood and whale bone found on beach ridges.

The age model, based on 13 C14 dates, is marred by “contamination” from a discrete, but unknown source of old remains of terrestrial (why?) plants. Eight dates from throughout the core apparently represent this material only, while five are considered to be “clean” with no contamination. This does shed some doubt on the model, and it would have been useful with some OSL and/or magnetic declination dates, but the authors give a fair discussion of the problems, and the model stands with its uncertainties as best solution to the problem.

We thank reviewer #5 for the assessment of our age model, and the way we discuss its limitations. We also like to take this opportunity to clarify the mentioned “contamination” (rather, reworking). As also detailed in our previous rebuttal, and in the core chronology

section of the manuscript, all these outliers are dated from terrestrial plant material and cluster around 10.5 ka BP. This age is consistent with chunks of peat, dated on the shores of adjacent Sørkapp Land as reported by Salvigsen and Elgersma (1993). Based on their sea-level tie-points, which envision a low stand around 5m below modern sea-level around this time (also see previous comment), the authors argue that this material is (and has been) wave-scoured from the (now submerged: 3-9 m deep) shallows between Sørkapp Land and Sørkappøya, which were peat lands at the time (so, around 10.5 ka BP). We hope this helps clarify this matter, and we have also made additional modifications to our “chronology” section to make sure this important point is conveyed with greater clarity.

In summary, the core gives important new evidence on Holocene storminess from a unique locality in the Arctic. It is well researched with state-of-the-art methods and a comprehensive survey of relevant literature from the North Atlantic coasts, claiming that the future along presently sea ice covered coasts may not be as stormy as anticipated by others. It should be of interest to a broad audience of climate researchers.

We once again thank reviewer #5 for a positive assessment of our work.

REVIEWERS' COMMENTS

In the text below, we *address* each comment in *italics*.

Reviewer #4 (Remarks to the Author):

I have read the rebuttal and checked the revised version of the manuscript. Overall, the authors have presented a much more honest representation of what the data does and does not show. The study still has weaknesses but can be published. Below (is) my evaluation of the concerns previously raised.

1) Previously, the study was presented as if the data could be used to reconstruct Arctic storminess. The authors now acknowledge that, actually, the study presents a record of summer windiness, during the time of the year when few storms occur. They acknowledge that the provided data may or may not be representative of changes in the broader region. They have made this very clear throughout manuscript, and adjusted their title. I appreciate these changes, as they have markedly improved the manuscript.

We thank reviewer 4 for this positive assessment of the implemented changes, and for raising this rather important nuance, that was indeed overlooked in previous versions.

2) With regards to site-specificity and extrapolation of the the results, the authors have now toned down their claims and acknowledge that the results may, or may not, represent broader changes in the region and that further research is required.

We thank reviewer 4 for this assessment, and look forward carrying out more work in the future on a transect of sites along the Svalbard coast.

3) My third comment was that more evidence should be provided to support the interpretation of the principal components, which the study heavily relies on. The authors are very confident in their interpretation of PC2, which is predominantly loaded by the CT values, yet there is no concrete evidence that links the CT values (or PC2 as a whole) to the westerlies. A convincing way in which the PCA analysis could have been supported would be by showing a comparison of the PC scores with modern/historical wind data. However, due to the disturbed core top the authors were not able to do this. Therefore, this analysis and interpretation remains a significant weakness of the study, but I suggest the study is published and the community can form its opinion. The authors have also showed that the the PCs are independent – this was a mistake on my part and my apologies for that. The authors did well by now converting their dataset into a standardized data format. They also point out that PCA can be used to tease out “hidden” signals in the data. Of course this is true –and a PC that captures 11% may or may not be ‘significant’ – but my point was that with every PC the risk of picking up noise does increase. Therefore, additional evidence should

be provided that a certain PC represents a certain physical phenomenon, i.e. that it is meaningful. Again, this is very important in the case of this study, since no comparison of the PCs with modern/observational data is provided.

We acknowledge that overlap with historical observations would have been optimal, and hope to be able to do so in future research. In addition, we would like to re-iterate that we do provide additional evidence linking PC 2 to the Westerlies. Notably, the notion that association of coarse-grained indicator Zr/K with this axis, in light of the exclusive presence of coarse-grained material towards the West of Steinbruvatnet (see the grain size analyses of Fig. S4 in particular), as well as the sensitivity of CT density values to changes in grain size variations (see i.e. reference 74). We hope this additional clarification has further allayed the concerns raised by reviewer 4, and are thankful for their consent to publication, so that the community can weigh in.

4) The concern of ice-rafting is resolved as the authors clarify that only pebbles are available for ice-rafting.

Thank you.

Reviewer #5 (Remarks to the Author):

Thanks for the revised ms and the very comprehensive discussions. I was glad to see that my main point - stressing that this is a record of summer storminess only - has been acted on, both in the title and in the text. I was surprised to see that another of my points - the role of varying snow and lake ice cover for sediment input into the lake - apparently has had little or no impact, but I am satisfied that this issue has now been addressed.

From reading the other reviewers' comments from fields that I am not really into, my feeling is that this ms is now ready for publication - I agree.

We thank reviewer 5 for this positive assessment, and for their appraisal of our line of reasoning regarding the (negligable) influence of varying snow and lake ice cover.